# FD-Align: Feature Discrimination Alignment for Fine-tuning Pre-Trained Models in Few-Shot Learning

**Kun Song**[1]    **Huimin Ma**[1,†]    **Bochao Zou**[1]    **Huishuai Zhang**[3]    **Weiran Huang**[2,†]

[1]SCCE, University of Science and Technology Beijing
[2]Qing Yuan Research Institute, SEIEE, Shanghai Jiao Tong University
[3]Microsoft Research Asia

songkun@xs.ustb.edu.cn, {mhmpub, zoubochao}@ustb.edu.cn, huzhang@microsoft.com, weiran.huang@outlook.com

## Abstract

Due to the limited availability of data, existing few-shot learning methods trained from scratch fail to achieve satisfactory performance. In contrast, large-scale pre-trained models such as CLIP demonstrate remarkable few-shot and zero-shot capabilities. To enhance the performance of pre-trained models for downstream tasks, fine-tuning the model on downstream data is frequently necessary. However, fine-tuning the pre-trained model leads to a decrease in its generalizability in the presence of distribution shift, while the limited number of samples in few-shot learning makes the model highly susceptible to overfitting. Consequently, existing methods for fine-tuning few-shot learning primarily focus on fine-tuning the model's classification head or introducing additional structure. In this paper, we introduce a fine-tuning approach termed Feature Discrimination Alignment (FD-Align). Our method aims to bolster the model's generalizability by preserving the consistency of spurious features across the fine-tuning process. Extensive experimental results validate the efficacy of our approach for both ID and OOD tasks. Once fine-tuned, the model can seamlessly integrate with existing methods, leading to performance improvements. Our code could be found in `https://github.com/skingorz/FD-Align`.

## 1   Introduction

The Contrastive Language-Image Pre-training model (CLIP) [1] represents a groundbreaking development in multi-modal deep learning. By utilizing contrastive learning, CLIP aligns visual and textual representations within a unified embedding space, and exhibits superior performance in a variety of downstream tasks, including image classification [2, 3], detection [4], and segmentation [5, 6], which is typically done by fully fine-tuning CLIP using downstream data. However, in many real-world scenarios, there is often an insufficient amount of labeled data available. Thus, fully fine-tuning will lead to overfitting and significantly diminishing the model performance.

To mitigate this few-shot challenge, we consider using a proxy dataset correlated to the downstream target dataset for fine-tuning CLIP, aiming to obtain a model that can efficiently generalize to the few-shot target task. Directly fully fine-tuning CLIP on the proxy dataset is not feasible, as the fine-tuned model may overfit to the proxy data or have worse out-of-distribution (OOD) generalization [7], limiting its performance on the target task. As an example illustrated in Figure 1, the fully fine-tuned CLIP tends to focus more on local regions and less on the foreground compared to the original CLIP.

---

[†]Corresponding authors.

37th Conference on Neural Information Processing Systems (NeurIPS 2023).

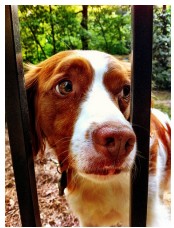 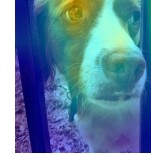 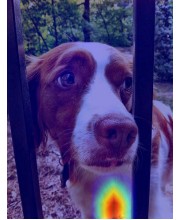 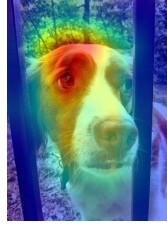

|  (a) Original Image | (b) CLIP | (c) Fully Fine-Tuned CLIP | (d) FD-Align |

Figure 1: (a) An image of a dog. (b) CLIP attention map, which pays more attention to the background in comparison with that to the dog. (c) The CLIP attention map after fully fine-tuning, which focuses more on locations with non-salient features. (d) The attention map after FD-Align fune-tuning, which tends to prioritize the dog's causal information while also paying attention to a minor portion of spurious information.

Such localized attention will weaken the model's robustness to spurious correlations [8], resulting in poor OOD generalization of the fully fine-tuned CLIP.

In this paper, our objective is to preserve the robustness of CLIP to spurious correlations during fine-tuning, i.e., its ability to distinguish between spurious and causal features. In particular, causal features represent the features related to the classes, while spurious features could be the features related to the context of the class. We want the fine-tuned CLIP can learn the causal features of the new classes while keeping the recognition ability for the spurious features. To achieve this, we propose a method of **F**eature **D**iscrimination **Align**ment (FD-Align). Specifically, we introduce a spurious feature classifier, ensuring that the classification probability distribution of spurious features remains consistent throughout the fine-tuning. Taking advantage of the potent alignment capabilities of CLIP's text and visual features, we utilize text features of category-agnostic descriptions (i.e., contexts) as spurious feature prototypes. The image features and spurious feature prototypes are subjected to similarity measurement to ascertain the current image's probability distribution over spurious features. By constraining the probability distribution of the image features extracted by the model before and after fine-tuning, we ensure consistency in the spurious features extracted by the model. Concurrently, while learning classification capabilities on the proxy dataset, the robustness of the model to spurious associations after fine-tuning is also ensured.

Our method maintains the model's robustness to spurious correlations while fine-tuning model on proxy dataset. As illustrated in Figure 1d, on the one hand, compared to CLIP, the fine-tuned model with FD-Align better focuses on the dog. On the other hand, compared to the localized attention of fully fine-tuning CLIP, FD-Align pays slight attention to some spurious information. This balance between attending to causal information and spurious information ensures the model's robustness to spurious correlations, thereby ensuring the model's OOD generalization. Extensive experiments validate the robust OOD performance of our approach, alongside the improvement of in-distribution (ID) performance. Furthermore, as shown in Figure 2, the model fine-tuned by FD-Align directly enhances the accuracy of existing methodologies without introducing additional inference cost.

Our paper makes the following contributions:

(1) We propose to use textual features to obtain spurious features of images;

(2) We propose a feature discrimination alignment fine-tuning architecture that ensures the OOD performance of the fine-tuned model by aligning spurious features extracted model before and after fine-tuning;

(3) Sufficient experiments demonstrate that our approach can significantly improve the performance of ID and OOD for few-shot learning, and it can improve the performance of existing methods without introducing additional training and inference costs.

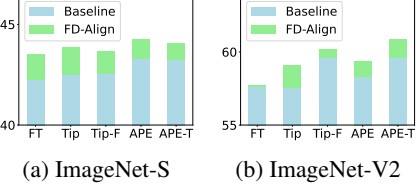

|  (a) ImageNet-S | (b) ImageNet-V2 |

Figure 2: Performance improvement for different datasets.

## 2 Related Work

**Few-Shot Learning.** The primary objective of few-shot learning is to train models of superior performance with a small number of samples. Prior methods primarily train models on base data and assess their performance on novel class data devoid of any shared categories. Approaches such as

MAML [9] adopt meta-learning to train a base learner on base data, followed by fine-tuning it on a limited amount of novel data to derive a model suited for novel data. ProtoNet [10] introduces the use of metric learning for training. Recently, a line of work start to introduce the modality of text into the few-shot image classification, such as AM3 [11], TRAML [12] and FILM [13]. All these models are trained from scratch. With the emergence of pre-trained bi-modal models such as CLIP, the acquisition of more precise image features is presently achievable by leveraging these pre-trained models. Consequently, the majority of current research focuses on how to use features extracted by CLIP to enhance the capability of few-shot learning. For instance, CoOp [14] and CoCoOp [15] model prompt's context words with learnable vectors, keeping all pre-trained parameters fixed. Tip-Adapter [3] and APE [16] do not require any backpropagation to train the adapter but create the weights through a key-value cache model constructed from the few-shot training set. VPT [17] introduces additional learnable parameters into the input space. However, all of these methods are processed with the backbone frozen, while our paper aims to further explore the possibility of fine-tuning the backbone itself. Although these methods show excellent performance, they do not exploit the potential of pre-trained models.

**Fine-Tuning of Pre-trained Model.** The most direct approach is fine-tuning the pre-trained model directly. However, fully fine-tuning decreases the OOD performance of the pre-trained model [7]. WiSE-FT [18] enhances performance by ensembling the weights of the zero-shot and fine-tuned models. Kumar et al. [7] first perform linear probing and then perform fully fine-tuning to ensure the OOD performance of the model. Xuhong et al. [19] introduce an additional regularizer to constrain the $l_2$ distance between zero-shot and fine-tuned CLIP. On the other hand, Mukhoti et al. [20] prevent the degradation of the foundational model's capacity by constraining image features extracted by zero-shot and fine-tuned CLIP in the feature space. Nevertheless, scant existing methods delve into the strategies for fine-tuning pre-trained models using limited data.

**Spurious Correlations.** Spurious correlations denote deceptive heuristics that hold for the majority of instances during training, but do not invariably apply [21]. For example, when the training instance involves a cow on grass, it is tempting to misconstrue the presence of grass as a causal determinant for categorizing cows. This misinterpretation, regarding grass as cows, embodies a form of spurious correlation. Eliminating spurious features generally leads to enhanced OOD performance; however, this can be accompanied by a decrease in ID performance [22]. Furthermore, even armed with a comprehensive understanding of the spurious feature, its extraction remains a non-trivial task [23]. The impact of spurious correlations persists within multimodal pre-trained models [24]. However, when model parameters and dataset sizes are large enough, Vision Transformer (ViT) exhibits increased robustness against spurious correlations [8]. Furthermore, the CLIP architecture enhances the robustness of the vision encoder to spurious correlations [25]. Consequently, improving robustness against spurious correlations can effectively maintain the OOD performance of the model.

## 3 The Proposed Method

### 3.1 Problem Definition

Suppose that we have a pre-trained CLIP [1] which contains a visual encoder $f_0$ and a text encoder $g_0$. In addition, we have access to a few-shot proxy dataset $\mathcal{D} \subset \mathcal{X} \times \mathcal{Y}$, where each class has very limited samples and each sample comprises an image $x$ and its corresponding label $y$. The objective is to fine-tune the pre-trained CLIP using this proxy dataset, aiming to enhance its zero-shot performance for unseen target tasks related to the proxy dataset.

### 3.2 Fine-Tuning on Proxy Dataset

We freeze CLIP's text encoder $g_0$ during fine-tuning and make the visual encoder learnable. First, we initialize the visual encoder $f_t$ using the parameters of the visual encoder of pre-trained CLIP $f_0$. The visual encoder $f_t$ is then used to extract the feature $f_t(x)$ of the image $x$. With the help of the remarkable text-vision alignment capability of CLIP, we use the textual features of the class names of each class as class prototypes. Following CLIP, for any class $y$, we combine $M$ prompt templates $(P_1, \ldots, P_M)$ with the class name and obtain $M$ prompts $[P_1, y], \ldots, [P_M, y]$. We then use the text encoder $g_0$ to extract the features of the above $M$ prompts. Subsequently, we calculated the means of

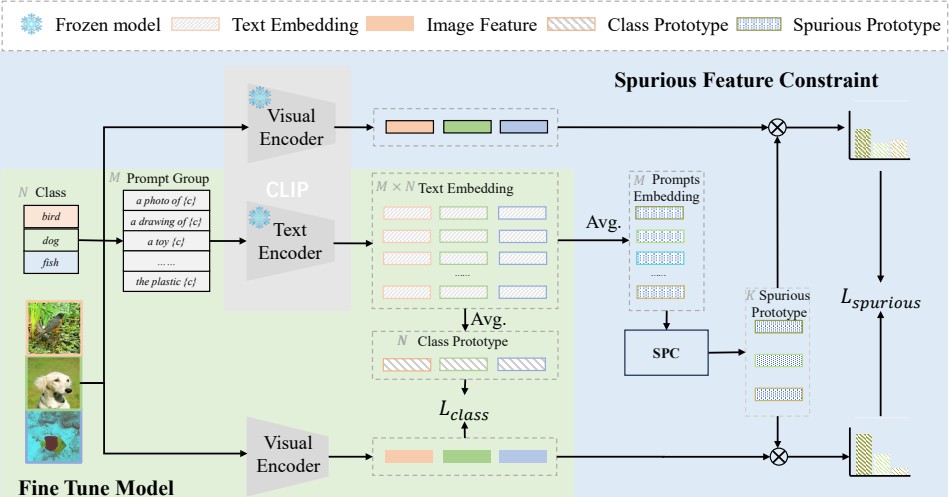

Figure 3: The class names and prompts are combined and inputted into the text encoder to obtain text embeddings. We calculate the mean separately in the prompt and class dimensions to derive the class prototype and prompt embedding. On the one hand, the image features are extracted using the fine-tuned visual encoder, and class distribution are calculated based on the class prototype to calculate the class loss. On the other hand, we use spurious prototype correction (SPC) module to correct the prompt embedding. By calculating the cosine similarity between the image features and the spurious prototype, we obtain the distribution over spurious features and calculate the spurious loss.

the $M$ features to obtain the prototype of the corresponding class, namely, the prototype of class $y$ is

$$\mu_y^{\text{class}} := \frac{1}{M} \sum_{j=1}^{M} g_0([P_j, y]).$$

We calculate the cosine similarity $s(\cdot, \cdot)$ between the image feature and the class prototypes and produce a distribution over classes for the image. Finally, the class loss is calculated using cross-entropy loss by

$$\mathcal{L}_{\text{class}} = -\frac{1}{|\mathcal{D}|} \sum_{(x_i, y_i) \in \mathcal{D}} \log \frac{\exp(s(f_t(x_i), \mu_{y_i}^{\text{class}}))}{\sum_{y \in \mathcal{Y}} \exp(s(f_t(x_i), \mu_y^{\text{class}}))}, \tag{1}$$

where $\mathcal{Y}$ is the label set.

### 3.3 Spurious Feature Constraint

Fully fine-tuning CLIP on the proxy data affects the robustness of the model to unseen data. To preserve the model's performance on out-of-distribution data during fine-tuning, we maintain the model's robustness to spurious correlations during fine-tuning. That is, keeping the spurious feature extracted by the model before and after fine-tuning unchanged. We first calculate the mean of the features of each prompt template $P_j$ over all classes as prototypes of the prompt template $P_j$, that is,

$$\mu_{P_j}^{\text{spurious}} := \frac{1}{|\mathcal{Y}|} \sum_{y \in \mathcal{Y}} g_0([P_j, y]).$$

We can calculate the similarity between the feature extracted by the fine-tuned model and the spurious prototypes and produce the distribution over spurious features as follows.

$$\mathcal{P}_{\text{spurious}}(x; f_t) = \text{SoftMax} \left[ s\left(f_t(x), \mu_{P_1}^{\text{spurious}}\right), \ldots, s\left(f_t(x), \mu_{P_M}^{\text{spurious}}\right) \right].$$

Similarly, we can use the pre-trained visual encoder $f_0$ to extract the feature and produce the distribution over spurious features as follows.

$$\mathcal{P}_{\text{spurious}}(x; f_0) = \text{SoftMax}\left[s\left(f_0(x), \mu_{P_1}^{\text{spurious}}\right), \ldots, s\left(f_0(x), \mu_{P_M}^{\text{spurious}}\right)\right].$$

We can ensure that the spurious features of the models before and after fine-tuning remain consistent by keeping the probability distributions of the models over spurious features consistent before and after fine-tuning, that is

$$\mathcal{L}_{\text{spurious}} = \frac{1}{|\mathcal{D}|} \sum_{(x_i, y_i) \in \mathcal{D}} \text{KL}\left(\mathcal{P}_{\text{spurious}}(x_i; f_t) \,||\, \mathcal{P}_{\text{spurious}}(x_i; f_0)\right). \qquad (2)$$

Finally, we optimize both (1) and (2) during fine-tuning to ensure classification ability and OOD robustness:

$$\mathcal{L}_{\text{total}} = \alpha \cdot \mathcal{L}_{\text{class}} + \beta \cdot \mathcal{L}_{\text{spurious}},$$

where we set $\alpha$ to 1 and $\beta$ to 20 in this paper.

### 3.4 Spurious Prototype Correction

Prompt templates are typically designed manually or generated by large-language models such as GPT. These templates often include redundant or illogical prompts. Consequently, the prototypes of spurious features calculated by these imprecise and redundant prompt templates lack accuracy. Therefore, filtering and processing these prototypes of spurious features is necessary.

Certain prompt templates may exhibit a lack of practical significance or irrationality, making them unsuitable for incorporation as spurious features, as exemplified by "itap of {class}." This scenario can result in the inaccuracy of spurious prototypes. To address this problem, we employ the Isolation Forest algorithm [26] to eliminate meaningless prototypes associated with spurious features, i.e., $\mu^{\text{spurious}} := \text{ISOLATIONFOREST}(\mu^{\text{spurious}}, n)$. We will retain the $n$ prototypes that exhibit the highest degree of rationality.

Moreover, there are cases where certain prompts exhibit excessive similarity. For example, prompts such as "a photo of the {class}.", "a photo of a {class}.", and "a photo of my {class}." show notable similarities. In such scenarios, a single piece of spurious information may correspond to multiple prompts. However, some spurious information aligns with only one prompt. Consequently, the relative weight of the probability of spurious information corresponding to a single prompt diminishes during the classification process. To address this issue, we employ the $k$-means algorithm to merge duplicate spurious features that arise from similar prompts, i.e., $\tilde{\mu}^{\text{spurious}} := k\text{-means}(\mu^{\text{spurious}}, k)$, where $k$ is the number of cluster centers.

## 4 Experiments

In this section, we will validate the OOD robustness of our approach. Furthermore, our method demonstrates promising results in ID data.

### 4.1 Setup

Throughout this section, if not otherwise specified, we use open source ViT-B/32 as the backbone of the CLIP, and the prompts we use are OpenAI ImageNet prompt templates[2]. we set $n$ to 60, $k$ to 20 in the spurious prototype correction stage. We employ the Stochastic Gradient Descent (SGD) optimizer for model fine-tuning, conducting the process over 60 epochs. Since little prior work on CLIP has been done to fine-tune the backbone for few-shot learning, we mainly compare our method with the fully fine-tuning strategy and WiSE-FT [18].

---

[2]https://github.com/openai/CLIP/blob/main/notebooks/Prompt_Engineering_for_ImageNet.ipynb

**Dataset**

For the OOD setting, we evaluate our method on two different datasets. On the one hand, we train the model on ImageNet [27] following the few-shot task in CLIP and test the performance on two OOD variants of ImageNet [27]: ImageNetV2 [28] and ImageNet-Sketch [29] with the same 1000 classes, On the other hand, we follow the traditional few-shot learning strategy and fine-tune the model on the train split of miniImageNet and evaluate the model on Meta-dataset [30], BSCDFSL benchmark [31] and DomainNet [32], for a total of 19 datasets. These datasets cover a wide range of image styles, such as natural images, satellite images, medical images, and sketch images.

For ID setting, we use the 11 image recognition datasets following CoOp [14], which contain ImageNet [27], StanfordCars [33], UCF101 [34], Caltech101 [35], Flowers102 [36], SUN397 [37], DTD [38], EuroSAT [39], FGVCAircraft [40], OxfordPets [41] and Food101 [42]. These datasets cover a wide range of different visual recognition tasks such as generic object classification, fine-grained classification, action, scene, texture, etc.

| Method | CLIP | Baselines | | | | | Baselines + FD-Align | | | | |
|---|---|---|---|---|---|---|---|---|---|---|---|
| | | FT | Tip | Tip-F | APE | APE-T | FT | Tip | Tip-F | APE | APE-T |
| ImageNet | 63.34 | 64.91 | 65.49 | 68.43 | 66.55 | 68.74 | **66.39** | **65.49** | **68.70** | **67.59** | **69.15** |
| ImageNetS | 42.31 | 42.24 | 42.48 | 42.54 | 43.28 | 43.23 | **43.50** | **43.84** | **43.67** | **44.23** | **44.04** |
| ImageNetV2 | 55,92 | 57.63 | 57.58 | 59.58 | 58.31 | 59.58 | **57.73** | **59.10** | **60.17** | **59.36** | **60.83** |

Table 1: OOD results. We fine-tune the model on 16-shot ImageNet and evaluate it on the variants of ImageNet. FT means fully fine-tuning.

## 4.2 Results under OOD Setting

In order to evaluate the robustness of our method in OOD data, we fine-tune the model on ImageNet in 16-shot settings and test the performance on the variants of ImageNet. As shown in Table 1, in contrast to fully fine-tuning CLIP, our approach yields a performance boost of up to 1.75% on ImageNetA. Additionally, we directly integrate our fine-tuned backbone into Tip-adapter [3] and APE [16], leading to a substantial improvement in their generalization performance. The aforementioned findings demonstrate that FD-Align improves model performance while ensuring robust generalization. Furthermore, the fine-tuned model can seamlessly integrate into existing methods without requiring additional fine-tuning.

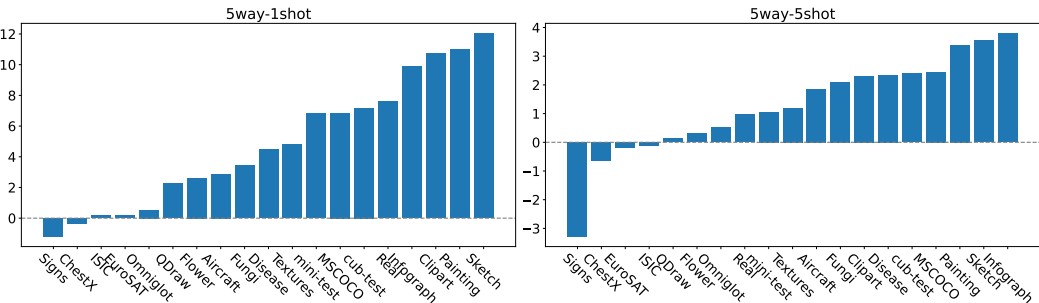

Figure 4: Performance improvement/decline of our method on different OOD datasets.

In addition, we also evaluate the robustness of our method in the $N$-way $K$-shot few-shot learning task. We fine-tune the model on the miniImageNet train split utilizing prototypical networks [10] and subsequently evaluate its performance on several different datasets. In our testing phase, we evaluate the precision of 2000 tasks for 5-way-1shot and 5-way-5shot scenarios in five distinct sets generated according to different random seeds. We then calculate the mean accuracy along with a confidence interval 95%. As illustrated in Table 2 and Figure 4, our approach yields notable performance enhancements across the majority of datasets. In particular, on Sketch dataset, our method achieves a performance improvement of 12.03% compared to CLIP. Nevertheless, on certain datasets, such as Traffic Sign, a marginal decrement in performance is observed with our approach. This may

| Datasets | 5way-1shot | | | 5way-5shot | | |
|---|---|---|---|---|---|---|
| | CLIP | WiSE-FT | FD-Align | CLIP | WiSE-FT | FD-Align |
| Mini-test [43] | 88.21±0.33 | 93.55±0.17 | **95.04±0.18** | 97.46±0.07 | 98.44±0.06 | **98.52±0.07** |
| CUB-test [44] | 75.21±0.78 | 81.16±0.71 | **82.38±0.69** | 91.48±0.34 | 93.41±0.32 | **93.87±0.24** |
| Textures [38] | 61.26±0.24 | 63.55±0.19 | **66.05±0.12** | 82.40±0.40 | 83.31±0.31 | **83.60±0.34** |
| Traffic Signs [45] | 58.51±0.11 | **60.84±0.29** | 57.32±0.26 | 76.67±0.18 | **78.11±0.24** | 73.39±0.29 |
| Aircraft [40] | 60.57±0.54 | 62.64±0.62 | **63.45±0.65** | 76.35±0.59 | 77.66±0.59 | **78.21±0.58** |
| Omniglot [46] | 83.27±0.37 | 83.56±0.28 | **83.81±0.25** | 94.29±0.13 | **95.26±0.09** | 94.81±0.19 |
| VGG Flower [36] | 90.88±0.31 | **94.16±0.23** | 93.50±0.24 | 98.65±0.10 | **99.06±0.09** | 98.95±0.09 |
| MSCOCO [47] | 62.30±0.38 | 67.28±0.32 | **69.16±0.28** | 78.93±0.38 | 81.08±0.35 | **81.37±0.24** |
| Quick Draw [48] | 62.22±0.61 | 62.54±0.59 | **64.49±0.58** | 82.65±0.31 | 82.78±0.37 | **82.78±0.28** |
| Fungi [49] | 50.42±0.32 | 53.10±0.27 | **53.83±0.30** | 71.59±0.18 | 73.28±0.10 | **73.69±0.14** |
| Plant Disease [50] | 70.64±0.28 | **75.66±0.33** | 75.13±0.33 | 89.50±0.24 | 91.78±0.31 | **91.84±0.19** |
| ISIC [31, 51] | 28.66±0.35 | **29.40±0.34** | 28.84±0.44 | 39.02±0.24 | **39.54±0.40** | 38.91±0.44 |
| EuroSAT [39] | 60.20±0.48 | **63.99±0.39** | 60.39±0.43 | 77.43±0.21 | **80.96±0.19** | 77.25±0.16 |
| ChestX [52] | **22.65±0.27** | 22.27±0.28 | 22.31±0.17 | **25.58±0.08** | 25.08±0.14 | 24.95±0.15 |
| Real [32] | 84.84±0.32 | 89.96±0.26 | **92.45±0.28** | 96.39±0.11 | 97.16±0.02 | **97.36±0.04** |
| Sketch [32] | 67.24±0.60 | 73.84±0.56 | **79.27±0.38** | 87.66±0.27 | 89.87±0.16 | **91.20±0.19** |
| Infograph [32] | 55.72±0.21 | 61.93±0.47 | **65.61±0.17** | 78.23±0.36 | 80.87±0.30 | **82.02±0.37** |
| Painting [32] | 68.05±0.18 | 74.92±0.33 | **79.06±0.32** | 87.99±0.21 | 90.26±0.22 | **91.37±0.21** |
| Clipart [32] | 75.14±0.12 | 81.55±0.26 | **85.86±0.21** | 92.52±0.09 | 94.06±0.13 | **94.83±0.11** |

Table 2: The performance of CLIP, WiSE-FT and FD-Align with SPC (FD-Align)

be attributed to the dataset images primarily encompassing the object of identification, devoid of substantial contextual misinformation that might adversely impact direct fine-tuning. Moreover, our method consistently outperforms WiSE-FT [18] in most datasets. In the few instances where we do not surpass its performance, our results remain comparable.

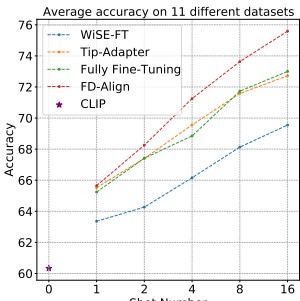

Figure 5: Performance comparison of different methods on 11 datasets.

| Methods | 1shot | 2shot | 4shot | 8shot | 16shot |
|---|---|---|---|---|---|
| Tip | 64.11 | 64.36 | 64.63 | 65.17 | 65.49 |
| Tip + **FD-Align** | 64.51 | 65.33 | 65.76 | 66.79 | 67.28 |
| Tip-F | 64.64 | 65.18 | 65.78 | 67.21 | 68.43 |
| Tip-F + **FD-Align** | 64.86 | 65.61 | 66.11 | 67.58 | 68.70 |
| APE | 65.36 | 65.69 | 66.00 | 66.55 | 66.55 |
| APE + **FD-Align** | 66.71 | 67.29 | 67.40 | 67.76 | 67.69 |
| APE-T | 65.89 | 66.18 | 66.82 | 67.99 | 68.74 |
| APE-T + **FD-Align** | 66.84 | 67.37 | 67.81 | 68.73 | 69.15 |

Table 3: Results on ImageNet using our backbone with different methods.

## 4.3 Results under ID Setting

In addition, we evaluate the performance of our method using ID data. Models were trained on various datasets using 1, 2, 4, 8, and 16 shots, respectively, and subsequently evaluate performance on the respective datasets. Figure 5 illustrates a comparative analysis of the average performance between our method and existing methods on 11 datasets. As depicted, our method significantly outperforms other strategies and yields increasingly substantial performance enhancements as the shot number increases. Previous research indicates that WiSE-FT effectively elevates the fine-tuning performance of models in zero-shot learning. However, in few-shot learning, relative to fully fine-tuning, employing WiSE-FT with a fusion parameter of 0.5 results in a performance notably inferior to comprehensive fine-tuning. Our analysis posits that, in few-shot learning, the paucity of fine-tuning samples results in minimal alterations in model fine-tuning. Consequently, following fusion, the model parameters more closely resemble those of CLIP, thereby influencing the model performance.

### 4.4 Discussion

#### 4.4.1 The Generality of FD-Align

To validate the generalizability of models fine-tuned via FD-Align across various methods, we directly incorporated the FD-Align fine-tuned visual encoder into existing approaches and evaluate ID performance. As shown in Table 3, the visual encoder fine-tuned with FD-Align markedly improves the performance of existing methods, achieving a peak improvement of 1.79%. The visual encoder fine-tuned with FD-Align, can be seamlessly transitioned into existing methods, bolstering performance without incurring additional costs.

| Methods | 1shot | 2shot | 4shot | 8shot | 16shot |
|---|---|---|---|---|---|
| CLIP | | | 60.33 | | |
| Fully Fine-tuning CLIP | 63.48 | 64.87 | 68.10 | 71.14 | 73.43 |
| FD-Align (80 templates) | 63.90 | 65.64 | 68.10 | 71.30 | 74.03 |
| FD-Align (Tip templates) | 61.14 | 62.39 | 63.37 | 60.34 | 66.30 |
| FD-Align + SPC | **63.92** | **65.68** | **68.63** | **71.66** | **74.38** |

Table 4: Ablation results. All methods use the same learning rate and calculate the average performance across 11 datasets. The setup of 80 templates means using all the features of 80 templates as spurious prototype. The setup of Tip templates means using the prompts templates proposed by Tip-Adapter, and SPC means spurious prototype correction.

#### 4.4.2 Why Remove Outliers and Cluster?

In this section, we analyze the importance of eliminating outliers from spurious features and the subsequent clustering of residual points via empirical analysis. As illustrated in Table 4, employing the Spurious Prototype Clustering (SPC) method yields enhanced performance for the spurious prototype relative to directly utilizing all 80 templates. This underscores the efficacy of removing outliers and redundant values from spurious features. Furthermore, we experimented with features derived from manually designed prompt templates as spurious prototypes. While Tip-adapter handpicked 7 templates, our findings indicate that leveraging these 7 templates directly as spurious prototypes precipitates a marked performance degradation. A closer examination revealed that the template "itap of a {class}" from Tip-adapter was identified and excluded as an outlier by SPC during its initial phase. This underscores the ability of SPC to autonomously filter outliers of spurious prototypes, thereby circumventing potential pitfalls inherent in manual selections.

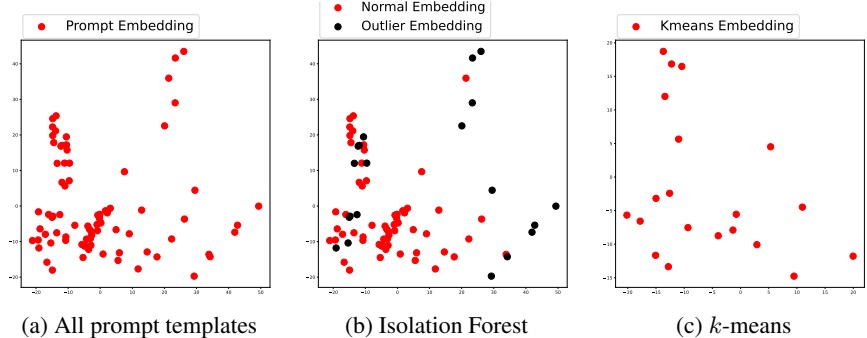

(a) All prompt templates        (b) Isolation Forest        (c) $k$-means

Figure 6: (a) represents all the features of prompt templates. (b) represents the features processed by Isolation Forest. (c) represents the final results clustered by $k$-means.

To more intuitively illustrate the impact of outlier removal and clustering, we visualize the spurious features. Figure 6a displays the features of all the prompt templates within the feature space, revealing the conspicuous presence of outliers and redundant values. Figure 6b shows the visualization after outlier elimination: red points represent preserved values, while black points denote deleted outliers, including "itap of a {class}". It is evident that numerous redundant values persist. As depicted in

Figure 6c, after the clustering process, the residual spurious features are uniformly distributed across various positions, thereby providing a rational representation of the spurious prototype.

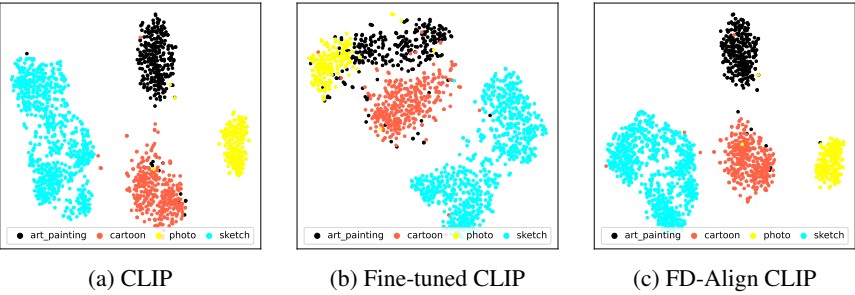

(a) CLIP            (b) Fine-tuned CLIP            (c) FD-Align CLIP

Figure 7: Feature visualization for the same class under different domains.

### 4.4.3 OOD Visualization

To more vividly demonstrate the influence of our method on model feature extraction, we conducted a visualization of the image features on PCAS [53], utilized for investigating OOD tasks, which contains category and domain annotations for images. Figure 7 illustrates the visualization of features for one of our categories. It is observable that CLIP adeptly distinguishes features across different domains. The ability of fully fine-tuned CLIP to differentiate information across various domains is comparatively diminished. After fine-tuning with FD-Align, the model reacquires the ability to discern information across different domains, substantiating FD-Align's ability to preserve the model's OOD performance. Additionally, as depicted in Figure 8, we visualize the features of different categories in various domains. On the sketch data, the model, after fully fine-tuning, exhibits a diminished capacity to differentiate between categories, whereas CLIP and FD-Align distinguish between various categories accurately.

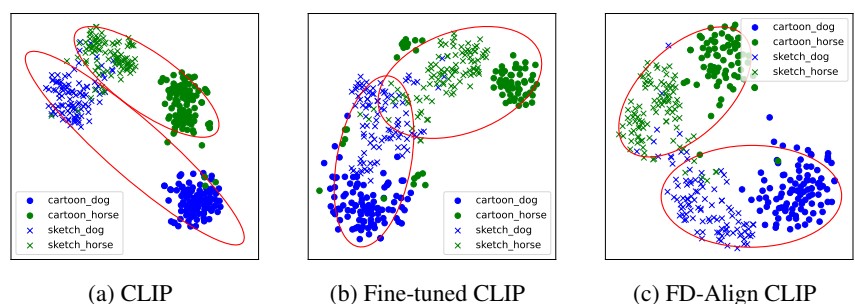

(a) CLIP            (b) Fine-tuned CLIP            (c) FD-Align CLIP

Figure 8: Feature visualization for different class under different domains.

### 4.4.4 Training Stability

Figure 9 depicts the evolution of model accuracy and loss on the validation set throughout the fully fine-tuning and FD-Align processes. Notably, during the initial phases of fine-tuning, fully fine-tuning prematurely encounters overfitting, culminating in a decline in accuracy and a surge in classification loss. Conversely, FD-Align adeptly maintains model robustness and stability throughout the fine-tuning process, effectively circumventing overfitting.

### 4.4.5 Limitation

As shown in Tables 2, our method is able to preserve the generalization of the model while fine-tuning the model. However, in certain specific cases, such as Traffic Sign, where the images primarily consist of target objects and lack category-independent information, the preserved pre-trained model's category-independent ability adversely impacts the performance on such data. Additionally, our method does not perform well on special scenarios, such as ChestX and EuroSAT, for which one possible reason is that the prompts we use do not contain the category-independent information of these data. Therefore, additional prompts may need to be designed for this type of data.

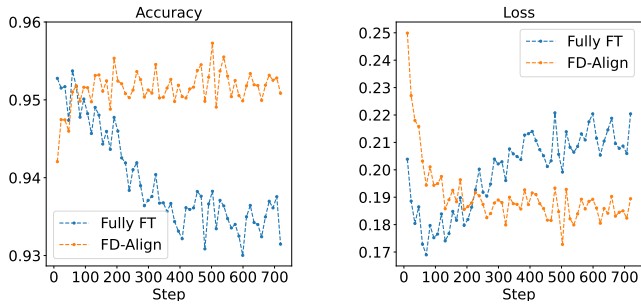

Figure 9: Validation accuracy and loss variation during the training process.

## 5    Conclusion

In this paper, we introduce a feature discrimination alignment fine-tuning (**FD-Align**) method pre-trained models in few-shot learning. Leveraging the remarkable text-visual alignment capabilities of CLIP, we employ text features of category-agnostic descriptions as spurious feature prototypes. Furthermore, we constrain the probability distribution of image features extracted by the model on spurious features, both before and after fine-tuning, to ensure the robustness of the model subsequent to fine-tuning. Experimental results substantiate the efficacy of our approach in enhancing fine-tuning performance while ensuring robustness across distributions.

## Acknowledgment

Huimin Ma and Bochao Zou are supported by the National Natural Science Foundation of China (No.U20B2062, No.62227801, No.62206015) and National Key Research and Development Program of China (2022ZD0117900). Weiran Huang is supported by 2023 CCF-Baidu Open Fund and Microsoft Research Asia.

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

# Appendix

## A  Detailed Results of Different ID Datasets

Figure 10 delineates the effectiveness of our approach across different datasets. Impressively, FD-Align surpasses both WiSE-FT and the fully fine-tuning on a vast majority of these datasets. As the shot count rises, the advantage of FD-Align becomes even more salient. It is pertinent to note that when data is scarce, WiSE-FT tends to underperform compared to fully fine-tune.

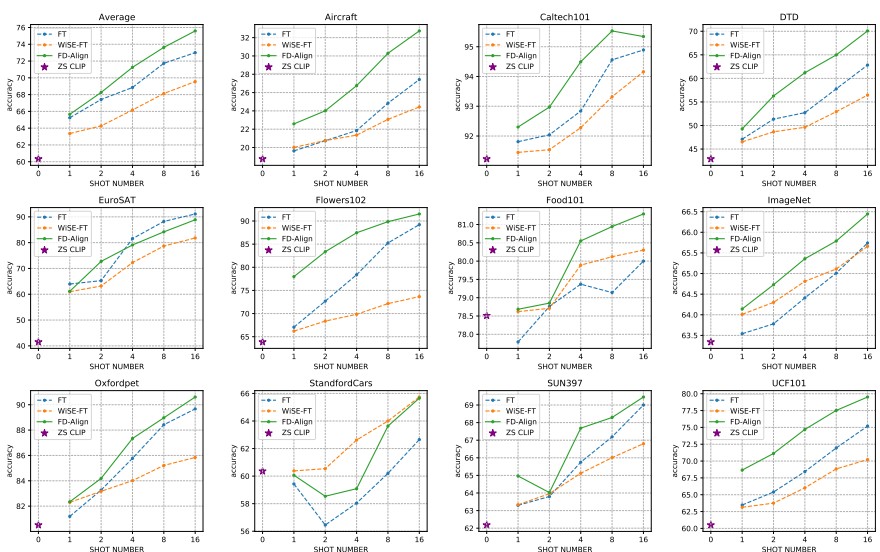

Figure 10: Performance comparison of different methods on 11 datasets.

## B  Time Cost of FD-Align

Table 5 shows the time cost of fully fine-tuning and FD-Align on 1-shot ImageNet. Evidently, FD-Align introduces only a negligible increase in time overhead. It is noteworthy that during the inference phase, both FD-Align and fully fine-tuning share identical processes, resulting in consistent inference cost. A significant advantage of FD-Align is its ability to boost performance across varied methodologies with just fine-tuning once.

| METHOD | Time |
|---|---|
| Fully Fine-tuning | 7min 9s |
| FD-Align | 8min 53s |

Table 5: Time cost of Fully Fine-tuning and FD-Align.

## C  More Results on OOD Datasets

We also evaluate the performance of fully fine-tuned models on the OOD dataset. All models are trained on miniImageNet and performance is tested on other datasets. As shown in Table 6, our method performs better on Meta-Dataset, but worse on BSCDFSL. After analysis, we believe that the reason is that the model is fine-tuned on miniImageNet, which mainly contains natural image information, so the model better retains the domain information of natural images. However,

BSCDFSL mainly contains satellite images, medical images, and other nonnatural images, and our model fine-tuning does not retain the domain information of these datasets. Furthermore, our model affects the classification ability in the classification process. Therefore, on such datasets, our method may be slightly degraded compared to the direct fine-tuning of CLIP.

| | Dataset | 5way-1shot | | | 5way-5shot | | |
|---|---|---|---|---|---|---|---|
| | | FT | WiSE-FT | FD-Align | FT | WiSE-FT | FD-Align |
| Meta-dataset | mini_test | 94.15±0.19 | 93.55±0.17 | **95.04±0.18** | 98.13±0.12 | 98.44±0.06 | **98.52±0.07** |
| | cub-test | 80.78±0.69 | 81.16±0.71 | **82.38±0.69** | 92.95±0.29 | 93.41±0.32 | **93.87±0.24** |
| | Textures | 64.77±0.12 | 63.55±0.19 | **66.05±0.12** | 82.57±0.46 | 83.31±0.31 | **83.60±0.34** |
| | Traffic Signs | **62.92±0.36** | 60.84±0.29 | 57.32±0.26 | 77.95±0.31 | **78.11±0.24** | 73.39±0.29 |
| | Aircraft | 63.44±0.69 | 62.64±0.62 | **63.45±0.65** | 77.66±0.52 | 77.66±0.59 | **78.21±0.58** |
| | Omniglot | 80.48±0.35 | 83.56±0.28 | **83.81±0.25** | 93.65±0.15 | **95.26±0.09** | 94.81±0.19 |
| | VGG Flower | 93.81±0.11 | **94.16±0.23** | 93.50±0.24 | 98.81±0.11 | **99.06±0.09** | 98.95±0.09 |
| | MSCOCO | **68.74±0.35** | 67.28±0.32 | 66.05±0.12 | 81.00±0.31 | 81.08±0.35 | **81.37±0.24** |
| | Quick Draw | 63.07±0.44 | 62.54±0.59 | **64.49±0.58** | 82.11±0.41 | 82.78±0.37 | **82.78±0.28** |
| | Fungi | **53.90±0.14** | 53.10±0.27 | 53.83±0.3 | 72.28±0.13 | 73.28±0.10 | **73.69±0.14** |
| BSCDFSL | Plant Disease | 73.99±0.35 | **75.66±0.33** | 75.13±0.33 | 90.27±0.38 | 91.78±0.31 | **91.84±0.19** |
| | ISIC | **29.66±0.44** | 29.40±0.34 | 28.84±0.44 | **40.27±0.39** | 39.54±0.40 | 38.91±0.44 |
| | EuroSAT | **64.49±0.34** | 63.99±0.39 | 60.39±0.43 | **81.14±0.25** | 80.96±0.19 | 77.25±0.16 |
| | ChestX | 21.76±0.24 | 22.27±0.28 | **22.31±0.17** | 23.88±0.17 | **25.08±0.14** | 24.95±0.15 |
| DomainNet | Real | **92.48±0.20** | 89.96±0.26 | 92.45±0.28 | 97.22±0.05 | 97.16±0.02 | **97.36±0.04** |
| | Sketch | 79.01±0.56 | 73.84±0.56 | **79.27±0.38** | 90.55±0.28 | 89.87±0.16 | **91.2±0.19** |
| | Infograph | 65.48±0.27 | 61.93±0.47 | **65.61±0.17** | 81.52±0.48 | 80.87±0.30 | **82.02±0.37** |
| | Painting | **79.34±0.31** | 74.92±0.33 | 79.06±0.32 | 90.98±0.16 | 90.26±0.22 | **91.37±0.21** |
| | Clipart | 85.84±0.45 | 81.55±0.26 | **85.86±0.21** | 94.46±0.09 | 94.06±0.13 | **94.83±0.11** |

Table 6: Results on OODdataset. FT stands for fully fine-tuning; WiSE-FT stands for performance after model fusion using Zero-Shot and fully fine-tuning; FD-Align stands for performance using our method.

