# OpenReview forum: "FD-Align: Feature Discrimination Alignment for Fine-tuning Pre-Trained Models in Few-Shot Learning"
_NeurIPS.cc/2023/Conference — NeurIPS 2023 poster_

### Official Review · Reviewer_TEuf · 2023-07-02

**Soundness:** 1 poor
**Presentation:** 3 good
**Contribution:** 1 poor
**Rating:** 3
**Confidence:** 5

**Summary:**

This work aims to adapt large pre-trained vision-language model to few-shot tasks. To this end, the authors propose to decouple the category-related and category-independent information to alleviate overfitting when adapting large model to few samples. They claim to maintain the visual encoder's ability to extract category-independent information during fine-tuning. They conduct experiments on various datasets, including CoOp-datasets and OOD datasets to verify the effectiveness of proposed methods.

**Strengths:**

1.The paper is organized and written well and the paper looks well polished. The overall story is clear to me.

2.The authors conduct experiments on extensive datasets for comprehensive evaluation.

**Weaknesses:**

1.The motivation is not very convincing. First, as shown in Figure 1 and line 36-42, I do not think the separability among domains could significantly affect the performance of downstream tasks, since we conduct classification within single domain. Second, as discussed in line 110-118, why we should contain the ability of extracting category-independent information. In fact, if the model can ignore the category-independent information and only extract the category-related information, it will obtain better performance and be very robust, even on different domain.

2.The proposed method is kind of naive and can not be considered as a systematic approach.

3.The experments are not very convicing, although they are conducted on various datasets. As shown in Table 1, some existing works are not compared, like CoOp, PLOT. I do not understand why this work use ViT-B as backbone, since many baselines use ResNet50. This work should conduct experiments with ResNet50 and compared with the reported results of baselines, instead of using a new backbone and reproduce the results baselines. Even with ViT-B, PLOT++ has achieve 70.6% average accuracy on 1-shot tasks, but this work only achieve 63.92%.

**Questions:**

Please answer the questions in weakness.

**Limitations:**

No.

---

> ### Author Rebuttal · Authors · 2023-08-09
>
> Q1 : Motivation is not very convincing.
>
> A1: Sorry for the confusion. Our primary objective focuses on **leveraging the pre-trained CLIP model for downstream few-shot tasks**. Specifically, when we have a dataset with limited samples, our aim is to enable CLIP to quickly adapt to it, and we also hope that the updated model can **generalize to other similar datasets**.
>
> The most straightforward approach is to fine-tune CLIP using downstream data. However, there exist two issues. First, since the downstream data is limited in the few-shot setting, directly fine-tuning CLIP with **insufficient data may lead to overfitting**, thereby diminishing the model's performance. Second, this direct fine-tuning might cause the** loss of the ability to distinguish between causal information and spurious information** in the original CLIP, which can negatively impact the updated model's out-of-domain (OOD) generalization.
>
> To address these two issues, we propose a controlled fine-tuning procedure for CLIP. Specifically, by imposing constraints that sustain its capability to differentiate between causal and spurious features, the updated model is steered away from both overfitting the few-shot data and fitting the spurious features.
>
> Q2: I do not think the separability among domains could significantly affect the performance of downstream tasks (Figure 1 and line 36-42).
>
> A2: Figure 1 of the paper visualizes the features of the same category under different domains. In order to better demonstrate the effectiveness of our motivation and the effect of disentanglement, we visualize the features of different categories under different domains. As shown in Figure 2 in the attached pdf of global response. We employ t-SNE dimensionality reduction to visualize distinct categories of image features within the contexts of both cartoon and sketch representations. This enables us to estimate the distribution of each category, delineated by an elliptic region. CLIP exhibits the capability to discern nuanced variations across diverse domains and categories. Subsequently, we fine-tuned the model on miniImageNet's training set. **On sketch images with large differences in domain from miniImageNet, the distinction of the feature extracted by fine-tunes model between the two categories deteriorates. And the distinction between domains becomes less obvious**. While our method can guarantee the differentiation between categories with larger differences and between domains.
>
> Q3: The proposed method is kind of naive and can not be considered as a systematic approach.
>
> A3: Thanks for the comments. While our approach may indeed seem simple on the surface, we view this **simplicity as one of its core strengths, rather than a limitation**. The ease of implementation and broad applicability that arise from this simplicity are valuable attributes in our opinion. We want to emphasize that **simplicity does not necessarily compromise innovation or the efficacy of a method**. In our particular case, this approach has demonstrated robust results, as substantiated by the experiments detailed in our submission.
>
> For additional context, we can draw a comparison to one of our key baselines, WiSE-FT. Despite employing what might be considered a straightforward weighted fusion strategy for subsequent classification, WiSE-FT has achieved remarkable success, even being recognized as a finalist for the prestigious CVPR2022 Best Paper award.
>
> We appreciate your insight and are open to further discussion if you have any specific suggestions or concerns that could enhance our work.
>
> Q4: Why not compared with some existing works.
>
> A4: Existing approaches mainly divide into two categories, one is to use additional adapters to adapt to the downstream task. The other is the learning of the prompt to improve the model performance. In contrast, our approach focuses on **improving the fine-tuning process of the CLIP backbone**. The fine-tuned backbone can be directly applied to previous methods to improve the performance of ID and OOD.
>
> We acknowledge the omission of pertinent comparisons with CoOp, Plot++, and other pertinent works. Your feedback is invaluable, and we are committed to rectifying this oversight in the revised rendition of our related work section.
>
> Furthermore, the results in **Table 2  and Table 3 in global response** prove the efficacy of our fine-tuned backbone. Experimental results demonstrate that our improved backbone can be seamlessly integrated into existing methods with noticeable performance gains.
>
> Q5: Why use ViT-B/32 as backbone rather than ResNet.
> A5: We need to fine-tune the backbone relative to the existing methods, so we choose ViT-B/32 as the backbone with the smallest memeroy in the CLIP open source backbone. Nevertheless, our approach is applicable under different backbone. We will add the performance of our method on ResNet50 in the revised version.
>
> Previous submissions suffered from inadequate hyperparameter tuning. In response, we meticulously re-tuned our hyperparameters to bolster performance, consequently yielding superior results. To provide a comprehensive perspective, we present the latest outcomes in the **Table 1 in global response**. Also, PLOT++ uses a stronger backbone ViT-B/16 relative to us.
>
> [1]  Robust fine-tuning of zero-shot models
>
> [2] Conditional Prompt Learning for Vision-Language Models
>
> [3] PLOT: Prompt Learning with Optimal Transport for Vision-Language Models

---

> > ### Author Response · Authors · 2023-08-15
> > **We would be grateful if you could take a look at the response**
> >
> > Dear Reviewer TEuf,
> >
> > Thank you for reviewing our paper. Just a friendly reminder that **the author-reviewer discussion will close soon**, and we eagerly await your feedback. In response to your comments, we've detailed our motivation to clear up any confusion and explained how our method differs from existing ones. To further validate our approach, we've added relevant experiments to the global response, even with space constraints. Could you please take a look at these updates?
> >
> > We're here to discuss any more questions or concerns you may have about our paper.
> >
> > With warm regards,
> >
> > Authors

---

> ### Comment · Reviewer_TEuf · 2023-08-19
> **The responses do not solve my concerns.**
>
> First, my concerns about motivation are not fully answered. Why distinguishing different domains is helpful for distinguishing different classes? Why the category-independent information is useful for classification? In fact, if the model can ignore all category-independent information and extract same features for same classes but different domains, it will perfectly classify different class, even for OOD samples.
>
> Second, the authors are suspected of deliberately using the backbone ViT-B/32 that is not commonly used in existing works to avoid direct comparison. The authors do not provide a convincing reason for not using ResNet50, so that it can really compare with most existing works. The authors also do not use ViT-B/16, which cannot be compared with PLOT++. Where are the results on ResNet50 and ViT-B/16?

---

> > ### Author Response · Authors · 2023-08-20
> > **A more detailed explanation of our motivation and results on other backbone.**
> >
> > Thank you for your reply to our feedback. We will provide further feedback on the motivation and performance  with other backbone.
> >
> > **Q1: A more detailed explanation of our motivation.**
> >
> > Ideally, it is desirable for feature to exhibit similarity when comparing data from the same category across many domains. CLIP demonstrates a strong ability to effectively differentiate between various categories and domains, and the features it extracts already have this property. Direct fine-tuning of the model will result in a better fit to the categories and domains of the fine-tuned data, at the corresponding cost of weaker recognition of domains not contained in the fine-tuned data. As in Figure 1 of our paper, the fine-tuned CLIP is less able to discriminate unseen domains; as shown in the sketch domain of Figure 2 in the PDF of the global rebuttal, CLIP is also less able to discriminate between different classes of data on unseen domains. This suggests that when fine-tuning CLIP, it also fine-tunes the features of domains not contained in the dataset. Since the dataset does not contain this type of data, fine-tuning for this type of feature is inaccurate, which leads to a weakening of the ability of the fine-tuned CLIP to recognize class-related features. In order to preserve CLIP's ability to recognize the same category, we need to not fine-tune features of unseen domains during fine-tuning. Therefore, we need to ensure that the fine-tuned CLIP is still able to distinguish data from different domains, so we need to ensure the ability of CLIP to distinguish category-irrelevant information during the fine-tuning process.
> >
> > **Q2: The results with other backbone.**
> >
> > To ensure a meaningful comparison with PLOT++, we conducted an evaluation of our approach on ImageNet dataset, employing the ViT-B/16 backbone. As shown in the  table, the results attributed to PLOT++ are sourced from its GitHub repository. Notably, our approach has achieved superior performance in specific shot numbers and demonstrates a superior average performance across all shot numbers. Subsequently, we will complement the results evaluated on other datasets in our revised paper.
> >
> > |  METHOD  |   1 shot  |   2 shot  | 4 shot | 8 shot | 16 shot | Average   |
> > |:--------:|:---------:|:---------:|:------:|:------:|:-------:|-----------|
> > |  PLOT++  | 66.45     | 68.28     | 70.40  | 71.31  | 72.60   | 69.80     |
> > | FD-Align | **69.24** | **69.65** | 70.28  | 71.02  | 71.60   | **70.36** |
> >
> > In addition, given that our paper introduces a simple and effective algorithm, presents comprehensive experiments, provides accessible code ensuring reproducibility, and raises no ethical concerns, **we believe it does not merit a reject status**, whose definition is
> >
> > >3: Reject: For instance, a paper with technical flaws, weak evaluation, inadequate reproducibility and incompletely addressed ethical considerations.

---

### Official Review · Reviewer_vJi7 · 2023-07-04

**Soundness:** 3 good
**Presentation:** 3 good
**Contribution:** 2 fair
**Rating:** 5
**Confidence:** 4

**Summary:**

This paper aims to enhance the performance of pre-trained CLIP for few-shot learning tasks while maintaining their generalizability and mitigating the risk of overfitting. To achieve this goal, the key contribution of this paper is the spurious information extractor that captures category-independent information from image features. By aligning the probability distributions of image features under the spurious weights, the fine-tuned model's ability to extract spurious features is preserved. Overall, this paper offers a robust CLIP fine-tuning method for improving the performance and generalization capabilities of pre-trained models in few-shot learning tasks.

**Strengths:**

1) Cross-domain and OOD few-shot learning are important research topics in this field. In this respect, the goal of the proposed method is backed by strong motivations.
2) Good performance. Fairly extensive experiments demonstrate that CLIP pretraining is promising for the few-shot and OOD few-shot learning tasks. Ablation studies indicate the innovations are helpful.

**Weaknesses:**

1) In my opinion, utilizing features from CLIP is knid of conflicting to few-shot settings. Sine CLIP is pretrained on multi-class large datasets, there could be some information leakage on similar classes, can it still be considered a few-shot learning task in this situation? As can be seen from Table 3, performance on some datasets that are unfamilar with pretrained CLIP (e.g., ChestX) are relatively poor.
2) More discussions/results regarding the cross-domain and dataset settings should be added. For instance, how about the performance on the same class under different domains (e.g., painted, sketch and real dog)? What kinds of pre-training source images (domain or object) are more helpful?
3) There is insufficient information about implementation details, such as the Isolation Forest and K-means algorithms, which makes it difficult to reproduce the work.
4) Apart from the training stability, it is not specified how long the proposed method takes for the entire training (i.e., pretraining and fine-tuning) and how efficient it is compared to existing studies.

**Questions:**

1) As mentioned in Weakness 1, the autors should give clearer explanations, or analyze the similarity between the pretraining dataset and the finetuning dataset.
2) For weakness 2, it is suggested to provide some cross-domain comparisons on the same class, several examples on the classification confidence are also helpful.
3) Typo errors such as ""wheile" -> "while" in line 260.

**Limitations:**

potential limitations have been discussed.

---

> ### Author Rebuttal · Authors · 2023-08-09
>
> Q1: Is using features from CLIP kind of conflicting to few-shot settings?
>
> A1: The consideration of CLIP features within the setting of few-shot learning remains a valid approach. Despite CLIP's extensive pretraining on vast datasets, it is important to acknowledge the potential for significant **deviations between its acquired knowledge and the specific requirements of downstream tasks**. Notably, recent advancements have witnessed the emergence of numerous CLIP-based few-shot learning paradigms, exemplified by methodologies such as Tip-adapter[1], APE[2], VPT[3], and others. These endeavors underscore the ongoing relevance and applicability of CLIP-based techniques in the context of few-shot learning scenarios.
>
> Q2: Why is performance on ChestX so poor?
>
> A2: The setting in Table 3 involves fine-tuning CLIP on miniImageNet's training set and subsequently evaluating its performance on the ChestX dataset, which predominantly consists of medical images. The distinct dissimilarity in overall data distribution and categories between ChestX and miniImageNet contributes significantly to the observed disparity in performance. Given CLIP's classification performance is still low, the discernible performance difference is attributable to the lack of pertinent information within CLIP's training corpus. Notably, medical images are **difficult to classify without a specialized background** due to the domain-specific challenges they pose. This observed performance discrepancy aligns with analogous findings in related research[4].
>
> Q3: Some cross domain comparisions on the same class.
>
> A3: In response to your inquiry, we performed a comprehensive evaluation regarding cross-domain and dataset settings. Specifically, we divided the few shot training set on imagenet and fine-tuned the model. We then performed performance tests on ImageNet and its variants datasets. The results of our study on 16 shots are shown in the table below, and our approach significantly improves cross-domain performance compared to the direct fine-tuning alternative. And the direct application of our fine-tuned backbone to existing methods can significantly improve performance.
>
> |           Method           | imageNet | imageNetA | imageNetR | imageNetS | imageNetV2 |
> |:--------------------------:|:--------:|:---------:|:---------:|:---------:|:----------:|
> |            CLIP            |   63.34  |   31.57   |   68.45   |   42.31   |    55.92   |
> |             FT             |   64.91  |   30.05   |    68.7   |   42.24   |    57.63   |
> |       FT+**FD-Align**      |   66.39  |    31.8   |    69.7   |    43.5   |    57.73   |
> |         Tip-adapter        |   65.49  |     -     |     -     |   42.48   |    57.58   |
> |  Tip-adapter+**FD-Align**  |   65.49  |     -     |     -     |   43.84   |    59.10   |
> |        Tip-adapter-F       |   68.43  |     -     |     -     |   42.54   |    59.58   |
> | Tip-adapter-F+**FD-Align** |   68.70  |     -     |     -     |   43.67   |    60.17   |
> |             APE            |   66.55  |     -     |     -     |   43.28   |    58.31   |
> |      APE+**FD-Align**      |   67.69  |     -     |     -     |   44.23   |    59.36   |
> |            APE-T           |   68.74  |     -     |     -     |   43.23   |    59.58   |
> |     APE-T+**FD-Align**     |   69.15  |     -     |     -     |   44.04   |    60.83   |
>
> Q4: The implementation details about Isolation Forest and K-means algorithms.
>
> A4: We have explained the details of our experiment in section 4.1. Specifically, 60 data points were retained in Isolation Forest and subsequently clustered into 20 classes using K-means. In addition, we have provided the **code in the supplement**. Again, we will subsequently **open source the code** to ensure the reproducibility of the method.
>
> Q5: The time cost of our method
>
> A5: In order to address this issue, we have compared the time required by our method compared to the direct fine-tuning method. The following table shows the results of 1 shot fine-tuning on imageNet. The results show that our method does not introduce much extra time consumption. Our method can be trained once and then applied directly to existing methods.  Moreover, our approach does not introduce any additional time overhead compared to CLIP during the inference phase.
>
> | Method       | Time       |
> |--------------|------------|
> | Fine-tuneing | 7min 9s  |
> | Ours         | 8min 53s |
>
> Q6: Some typos of this paper.
>
> A6: Thanks for pointing out the problem, we will fix it in our latest paper.
>
> [1] Tip-Adapter: Training-free CLIP-Adapter for Better Vision-Language Modeling
>
> [2] Not All Features Matter: Enhancing Few-shot CLIP with Adaptive Prior Refinement
>
> [3] Visual Prompt Tuning
>
> [4] Channel Importance Matters in Few-Shot Image Classification

---

### Official Review · Reviewer_vxTm · 2023-07-06

**Soundness:** 2 fair
**Presentation:** 2 fair
**Contribution:** 2 fair
**Rating:** 4
**Confidence:** 5

**Summary:**

This paper introduces a novel approach to tackle few-shot learning utilizing the powerful pretrained CLIP model. The primary objective is to construct multiple CLIP text prompts with different context and construct a term to regularize the learning process. This helps prevent overfitting to irrelevant correlations. Specifically, the method involves creating specific prompts for each class, combining category-independent factors such as "toy," "drawing," and "plastic," with the object category name. A loss term is designed to ensure that the extraction of spurious information remains consistent both before and after fine-tuning the CLIP model.

**Strengths:**

1. The notion of using the capabilities of the CLIP model to enumerate potential spurious factors, and subsequently constructing a regularization mechanism for few-shot learning, is a intriguing concept.

2. Some experimental results presented in this paper, particularly the findings depicted in Figure 4 and Figure 6, seems to be promising and somehow validates the proposed method.

**Weaknesses:**

1. The basic assumption of this paper, that if we fine-tuning CLIP model, the model tends to over-fit to the spurious information, needs a further validation. For example, it might be possible to use GradCAM or other visualization method to showcase such a phenomenon.
2. The notation is somewhat confusing and a careful proof-reading is needed as there are many typos in this paper, for example:
    (1) in line 134, The definition of C should be mentioned.
    (2) the index i is used twice in the equation after line 142, this causes confusion
    (3) W_spu was mistakenly written as W_sup in the same equation
    (4) The prompts in Figure 3 give an good example of category-independent information mentioned in line 142. It is better to refer to Figure3 in line 142 as a concrete example.
    (5) It is better to choose a different notation to denote object class, and context class, e.g., toy, plastic, ... in the equations
3. What is Fine-tuning CLIP baseline in Table 2? Is it using the first loss term only? If yes, the proposed method only leads to marginal improvement over this simple baseline. If that is the case, then the benefit of the proposed strategy might not be significant.
4. Also, the improvement by using IF and K-means is also marginal. In most cases, the improvement is less than 0.5%

**Questions:**

1. What is Fine-tuning CLIP baseline in Table 2? Is it using the first loss term only?


**Limitations:**

Yes, the limitation has been properly addressed.

---

> ### Author Rebuttal · Authors · 2023-08-09
>
> Q1: The validation of the basic assumption.
>
> A1: It is important to point out that our basic assumption is that **in few-shot learning, due to insufficient sample size to fine-tune the model, the model is prone to overfitting into the causal and spurious information it currently sees**. That is, the model may **incorrectly treat spurious information in the training sample as causal information of the current category**, thus losing the ability to distinguish between different spurious information. Our hypothesis can be demonstrated quite intuitively in Figure 1 in the paper. Figure 1 shows the t-SNE dimensionality reduction visualization of features extracted by different models for the same category over different domains. Compared to CLIP, the fine-tuned model is less capable of recognizing variance on data from different domains.
>
> Q2: The notation is somewhat confusing and careful proof-reading is needed as there are many typos in this paper.
>
> A2: Thanks for your suggestions, we will update them in later version.
>
> Q3: What is Fine-tuning CLIP baseline in Table 2? Is it using the first loss term only?
>
> A3: Yes, it is using the first term only. Previous submissions suffered from inadequate hyperparameter tuning. In response, we meticulously re-tuned our hyperparameters to bolster performance, consequently yielding superior results. To provide a comprehensive perspective, we present the latest results in the subsequent table.
>
> |      METHOD     |   1 shot  |   2 shot  |   4 shot  |   8 shot  |  16 shot  |
> |:---------------:|:---------:|:---------:|:---------:|:---------:|:---------:|
> |   CLIP(60.33)   |           |           |           |           |           |
> |     LP-CLIP     |   22.17   |   31.90   |   41.20   |   49.52   |   56.13   |
> |     WiSE-FT     |   63.36   |   64.27   |   66.16   |   68.13   |   69.55   |
> |   Tip-Adapter   |   65.52   |   67.40   |   69.56   |   71.58   |   72.72   |
> |        FT       |   65.24   |   67.42   |   69.85   |   71.73   |   73.00   |
> | **FT+FD-Align** | **65.65** | **68.26** | **71.25** | **73.63** | **75.59** |
>
>
> Q4: The improvement by using IF and K-means is marginal.
>
> A4: The utilization of IF aims to eliminate inappropriate prompts from the prompt group and mitigate the influence of redundant prompts. This strategy proves especially beneficial when the prompt group's quality is suboptimal. Given that we employ the prompts groups of CLIP for ImageNet, characterized by a paucity of inappropriate prompts, the performance enhancement is comparatively modest. Nevertheless, a performance gain is still attained. Furthermore, it is noteworthy that this procedure takes place during the training initialization phase, thereby exerting no impact on training velocity. Conversely, a reduction in the quantity of spurious factors contributes slightly to the acceleration of the training process.

---

> > ### Author Response · Authors · 2023-08-15
> > **We would be grateful if you could take a look at the response**
> >
> > Dear Reviewer vxTm,
> >
> > Thank you for reviewing our paper. Just a friendly reminder that **the author-reviewer discussion will close soon**, and we eagerly await your feedback. In response to your comments, we've made updates including a clearer t-SNE visualization to illustrate our assumption, an explanation of our choice to use Isolation Forest and Kmeans over manual removal, and adjustments to the hyperparameters to enhance performance. Our method now shows marked improvement over direct fine-tuning. Could you please take a look at these updates?
> >
> > We're here to discuss any more questions or concerns you may have about our paper.
> >
> > With warm regards,
> >
> > Authors

---

> > > ### Comment · Reviewer_vxTm · 2023-08-15
> > >
> > > I appreciate the author's dedication to providing further clarification and incorporating additional experimental findings. Following a thorough review of the rebuttal and the newly presented experimental outcomes, I continue to perceive limitations in both the performance and novelty of the proposed method. As such, I will uphold my original rating.

---

> > > > ### Author Response · Authors · 2023-08-15
> > > > **The novelty and performance of our method**
> > > >
> > > > Thank you for your reply to our feedback. We will provide further feedback on the novelty and performance of our method.
> > > >
> > > > **Novelty**
> > > >
> > > > In of few-shot learning, an excessive number of learnable parameters increases the risk of overfitting, a consequence of limited data. Consequently, prevailing approaches primarily direct their efforts towards enhancing performance while keeping the backbone unchanged. Nevertheless, the latent capacity within the model's backbone remains underutilized. Addressing this gap, our study centers on how to fine-tune the backbone, aiming to bolster its robustness without overfitting. Our backbone can be seamlessly replaced into existing methods and effectively improve performance.
> > > >
> > > > **Performance**
> > > >
> > > > In response, we have made revisions to the fine-tuning parameters of our methodology. The updated fine-tuning hyperparameters yield enhanced performance for both direct fine-tuning and our proposed technique compared to the previous submission. Furthermore, the revised findings demonstrate a notably enhanced performance in comparison to the approach of direct fine-tuning. Our proposed methodology demonstrates **2.59% improvement** when evaluated on 16 shots.
> > > >
> > > > |      METHOD     |   1 shot  |   2 shot  |   4 shot  |   8 shot  |  16 shot  |
> > > > |:---------------:|:---------:|:---------:|:---------:|:---------:|:---------:|
> > > > |   CLIP(60.33)   |           |           |           |           |           |
> > > > |   LP-CLIP [1]   |   22.17   |   31.90   |   41.20   |   49.52   |   56.13   |
> > > > |   WiSE-FT [2]   |   63.36   |   64.27   |   66.16   |   68.13   |   69.55   |
> > > > | Tip-Adapter [3] |   65.52   |   67.40   |   69.56   |   71.58   |   72.72   |
> > > > |        FT       |   65.24   |   67.42   |   69.85   |   71.73   |   73.00   |
> > > > | **FT+FD-Align** | **65.65** | **68.26** | **71.25** | **73.63** | **75.59** |
> > > >
> > > > Moreover, in order to assess the efficacy of our objective, we directly replace the backbone of existing methods, such as Tip and APE, with our fine-tuned backbone. The results indicate that our proposed strategy yields an **incremental improvement of up to 1.79%** on the ImageNet dataset.
> > > >
> > > > | METHOD                     | 1shot | 2shot | 4shot | 8shot | 16shot |
> > > > |----------------------------|-------|-------|-------|-------|--------|
> > > > | Tip-Adapter[1]             | 64.11 | 64.36 | 64.63 | 65.17 | 65.49  |
> > > > | Tip-Adapter+**FD-Align**   | 64.51 | 65.33 | 65.76 | 66.79 | 67.28  |
> > > > | Tip-Adapter-F[1]           | 64.64 | 65.18 | 65.78 | 67.21 | 68.43  |
> > > > | Tip-Adapter-F+**FD-Align** | 64.86 | 65.61 | 66.11 | 67.58 | 68.70  |
> > > > | APE[2]                     | 65.36 | 65.69 | 66.00 | 66.55 | 66.55  |
> > > > | APE+**FD-Align**           | 66.71 | 67.29 | 67.40 | 67.76 | 67.69  |
> > > > | APE-T[2]                   | 65.89 | 66.18 | 66.82 | 67.99 | 68.74  |
> > > > | APE-T+**FD-Align**         | 66.84 | 67.37 | 67.81 | 68.73 | 69.15  |
> > > >
> > > > Additionally, we conducted performance evaluations on some variant datasets of ImageNet. Our findings indicate that the utilization of our fine-tuned backbone resulted in current approaches achieving an **improvement of up to 1.52%** on the out-of-distribution (OOD) assignment. The aforementioned experimental findings provide comprehensive evidence that our approach yields superior performance and generalizability.
> > > >
> > > > | Method                     | imageNet | imageNetA | imageNetR | imageNetS | imageNetV2 |
> > > > |----------------------------|----------|-----------|-----------|-----------|------------|
> > > > | CLIP                       | 63.34    | 31.57     | 68.45     | 42.31     | 55.92      |
> > > > | FT                         | 64.91    | 30.05     | 68.7      | 42.24     | 57.63      |
> > > > | FT+**FD-Align**            | 66.39    | 31.8      | 69.7      | 43.5      | 57.73      |
> > > > | Tip-adapter                | 65.49    | -         | -         | 42.48     | 57.58      |
> > > > | Tip-adapter+**FD-Align**   | 65.49    | -         | -         | 43.84     | 59.10      |
> > > > | Tip-adapter-F              | 68.43    | -         | -         | 42.54     | 59.58      |
> > > > | Tip-adapter-F+**FD-Align** | 68.70    | -         | -         | 43.67     | 60.17      |
> > > > | APE                        | 66.55    | -         | -         | 43.28     | 58.31      |
> > > > | APE+**FD-Align**           | 67.69    | -         | -         | 44.23     | 59.36      |
> > > > | APE-T                      | 68.74    | -         | -         | 43.23     | 59.58      |
> > > > | APE-T+**FD-Align**         | 69.15    | -         | -         | 44.04     | 60.83      |
> > > >
> > > > The above is our response regarding novelty and performance. If you have any further questions, we look forward to discussing them with you.
> > > >
> > > > Once again, thank you for your valuable feedback and the opportunity to address your concerns. We eagerly anticipate any additional insights you may provide.
> > > >
> > > > [1] Tip-Adapter: Training-free CLIP-Adapter for Better Vision-Language Modeling. ECCV 2022
> > > >
> > > > [2] Not All Features Matter: Enhancing Few-shot CLIP with Adaptive Prior Refinement. ICCV 2023

---

### Official Review · Reviewer_ePYg · 2023-07-07

**Soundness:** 1 poor
**Presentation:** 2 fair
**Contribution:** 1 poor
**Rating:** 3
**Confidence:** 4

**Summary:**

This paper studies the problem of fine-tuning a pre-trained CLIP to downstream classification tasks with few-shot samples. The authors propose to fine-tune the category-dependent feature while retain the category-independent feature in order to improve the robustness of the fine-tuned model. The proposed method is compared with several baselines on several benchmarks.

**Strengths:**

The motivation is clear. The paper is clearly written.

**Weaknesses:**

- I'm not fully convinced by several key statements/arguments in the paper:
    - L114-118, why retaining the spurious features can help model robustness? I think the less spurious feature it learns, the better it transfers to different domain? I also do not understand "This allows the model to combine different spurious information based on the category information, thereby maintaining generalization when the object appears in a new context, style, or domain."
    - L131-132, "the object represents the causal information and 'a photo of' represents the spurious information". I do not understand why "a photo of" can contain spurious information? As the authors explain, the spurious information includes domain, style, or background informaiton. However, "a photo of" contains neither of them. It does not discriminate between different domains or backgrounds or styles. Then the spurious feature extractor is not actually extracting any spurious informaiton.

- Only fully fine-tuning and wise-ft are compared. What about other parameter-efficient fine-tuning methods such as LoRA [1] and VPT [2]? Also the gap between fully fine-tuning and FD-Align seems not really significant.

**Questions:**

See Weaknesses.

---

> ### Author Rebuttal · Authors · 2023-08-09
>
> Q1: Why retaining spurious features can help model robustness?
>
> A1: Sorry for the confusion. In fact, our objective is not to preserve spurious features of an image, but to **preserve the ability of CLIP to distinguish spurious features**. Illustrated in Figure 1a, CLIP demonstrates the capability to differentiate task-irrelevant information (deemed as spurious information within the classification task). Regrettably, direct fine-tuning destroys this capability, resulting in a model's inability to differentiate images affected by distributional bias.
>
> In few-shot tasks, during the fine-tuning process, the classifier encounters a combination of causal and spurious information within the features. Due to insufficient data, the fine-tuning procedure can readily lead to overfitting, causing the classifier to incorporate spurious features into its decision-making framework. This phenomenon leads to a decrease in the model's ability to recognize spurious information and thus errors in the testing phase.However, we impose constraints on the training process to preserve the model's ability to distinguish spurious features. At this point, the model learns mainly task-relevant information during the fine-tuning process. It refrains from overfitting spurious features onto the prevailing category attributes. Consequently, this safeguard guarantees that the model avoids overfitting to spurious information.
>
> Q2: How to understand "This allows the model to combine different spurious information based on the category information, thereby maintaining generalization when the object appears in a new context, style, or domain."?
>
> A2: To facilitate comprehension, we employ the object recognition task as an illustrative example. Within the context of the target recognition task, the object represents our task-relevant information, while the scene embodies our task-irrelevant information, **coexisting simultaneously**. Owing to the limited volume of training samples inherent in few-shot learning. Moreover, direct fine-tuning often leads to pronounced overfitting within the scene information abstraction, thereby compromising the model's capacity to differentiate between spurious and causal information. Consequently, during instances where the current category emerges within an unseen scene, the model's ability to discern objects within said scene may falter, consequently yielding classification errors. Notably, our proposed methodology ensures the retention of the model's discriminative acumen between causal and spurious information. In scenarios where objects manifest in previously unseen scenes, our model adeptly discriminates between object-specific and scene-specific information, consequently enhancing its proficiency in recognizing scenes absent from the training dataset. This, in turn, augments the overall generalization.
>
>
> Q3: Why does "a photo of" contain spurious information?
>
> A3: "a photo of" can be regared as a **style information**. Compared to sketch and painting, photo has a distinctly different style. Therefore, it is also a kind of spurious information.
>
> Q4: Why not compared with parameter-efficient fine-tuning methods?
>
> A4: Our goal is to fine-tune a better performing, more generalizable backbone for fine-tuning, so that it can replace the existing method's backbone and improve performance. In the following table, we provide comparative performance results on imagenet for replacing our backbone with the existing method.
>
> |METHOD|1shot|2shot|4shot|8shot|16shot|
> |-|-|-|-|-|-|
> |Tip-Adapter[3]|64.11|64.36|64.63|65.17|65.49|
> |Tip-Adapter+**FD-Align**|64.51|65.33|65.76|66.79|67.28|
> |Tip-Adapter-F[3]|64.64|65.18|65.78|67.21|68.43|
> |Tip-Adapter-F+**FD-Align**|64.86|65.61|66.11|67.58|68.70|
> |APE[4]|65.36|65.69|66.00|66.55|66.55|
> |APE+**FD-Align**|66.71|67.29|67.40|67.76|67.69|
> |APE-T[4]|65.89|66.18|66.82|67.99|68.74|
> |APE-T+**FD-Align**|66.84|67.37|67.81|68.73|69.15|
>
> Q5: The gap between fully fine-tuning and FD-Align seems not really significant.
>
> A5: Thanks for bringing this to our attention. Upon careful examination, we determined that the minor gap of method compared to fully fine-tuning was due to inadequate hyperparameter tuning. In fact, after re-tuning the hyperparameters of our method, a significant enhancement in performance can be observed. The updated results are shown in the table below.
>
> |      METHOD     |   1 shot  |   2 shot  |   4 shot  |   8 shot  |  16 shot  |
> |:---------------:|:---------:|:---------:|:---------:|:---------:|:---------:|
> |   CLIP(60.33)   |           |           |           |           |           |
> |     LP-CLIP     |   22.17   |   31.90   |   41.20   |   49.52   |   56.13   |
> |     WiSE-FT     |   63.36   |   64.27   |   66.16   |   68.13   |   69.55   |
> |   Tip-Adapter   |   65.52   |   67.40   |   69.56   |   71.58   |   72.72   |
> |        FT       |   65.24   |   67.42   |   69.85   |   71.73   |   73.00   |
> | **FT+FD-Align** | **65.65** | **68.26** | **71.25** | **73.63** | **75.59** |

---

> > ### Author Response · Authors · 2023-08-15
> > **We would be grateful if you could take a look at the response**
> >
> > Dear Reviewer ePYg,
> >
> > Thank you for reviewing our paper. Just a friendly reminder that **the author-reviewer discussion will close soon**, and we eagerly await your feedback. In response to some misunderstandings about the spurious feature, we've clarified it, hoping to address your concerns. We've also detailed how our method differs from the parameter-efficient fine-tuning method and included an experiment to prove our approach's effectiveness. Could you please take a look at these updates?
> >
> > We're here to discuss any more questions or concerns you may have about our paper.
> >
> > With warm regards,
> >
> > Authors

---

> > ### Comment · Reviewer_ePYg · 2023-08-19
> > **The concern remains for Q1**
> >
> > I do not quite understand why "preserve the ability of CLIP to distinguish spurious features" can avoid overfitting on spurious features? Say for a task of recognizing cows, we have training set of 5 images, each with a cow on the meadow, and the test set is a cow on the desert. If the model can recognize spurious features (meadow/desert), then the model might learn to classify based on the spurious features (meadow/desert) instead of causal features (cows). In contrast, if the model cannot recognize the spurious features, it can only classify based on the causal features, which is more robust. Therefore, without "the ability to distinguish spurious features", the model is more robust. Please correct me if I'm wrong.
> >
> > In the rebuttal, the authors say "...causing the classifier to incorporate spurious features into its decision-making framework. This phenomenon leads to a decrease in the model's ability to recognize spurious information...". This seems self-contradictory. Why learning to make decisions based on spurious features will harm the ability to recognize spurious features?

---

> > > ### Author Response · Authors · 2023-08-20
> > > **A more detailed explanation of our motivation with examples**
> > >
> > > Thank you for your feedback. We will explain our motivation in detail with example.
> > >
> > > In classification tasks, the most robust way is to have the model accurately recognize semantic information about the target object. It is well known that in few-shot learning, it is prone to use a spurious feature as a casual feature. e.g., when the training set is cows in the meadow and camels in the desert and the test sample is cows in the desert, it is likely to classify the cows in the desert as camels. To solve this problem, one way is to learn more robust semantic information of the target objects (features of cows and camels); the other is to enable the model to distinguish spurious information (meadow and desert) in the dataset without using it as a basis for classification. Fortunately, CLIP is able to robustly distinguish object features and does a good job of distinguishing different spurious features. Directly fine-tuning CLIP makes it learn features of the dataset better (cows in the grass and camels in the desert), but at the cost that the features of the current category that appear in other scenarios are also fine-tuned (cows in the desert), thus destroying the ability of CLIP to correctly discriminate data that is not contained in the dataset (cows in the desert). As shown in Figure 1 in our paper, direct fine-tuning of CLIP destroys its ability to distinguish between domains that do not exist in the dataset. As shown in Figure 2 of the pdf of global rebuttal, direct fine-tuning of CLIP on imageNet corresponds to a diminished ability to distinguish features of different classes of objects on sketch. This suggests that fine-tuning destroys the ability of CLIP to distinguish spurious information that is not contained in the dataset, which leads to a weaker ability to recognize targets in this spurious information. Therefore, we need to ensure that during fine-tuning, CLIP only fine-tunes the features of the data contained in the dataset (cows in the grass, camels in the desert) and does not change the features of the spurious information that is not contained (cows in the desert). So we need to constrain the training process during fine-tuning so that it retains the ability to distinguish spurious information.
> > >
> > > Furthermore, retaining the ability of the model to distinguish spurious information and not using spurious information as a basis for classification are not contradictory. If the model does not have the ability to distinguish spurious information, for example, a cow in the desert, it is likely to use the desert as causal information and thus classify it as a camel. When the model has the ability to distinguish between spurious information, it can distinguish between cows in the desert, where the desert is spurious information and the cows are causal information, and thus be able to classify it correctly.
> > >
> > > In addition, given that our paper introduces a simple and effective algorithm, presents comprehensive experiments, provides accessible code ensuring reproducibility, and raises no ethical concerns, **we believe it does not merit a reject status**, whose definition is
> > >
> > > >3: Reject: For instance, a paper with technical flaws, weak evaluation, inadequate reproducibility and incompletely addressed ethical considerations.

---

### Official Review · Reviewer_EKx5 · 2023-07-25

**Soundness:** 2 fair
**Presentation:** 2 fair
**Contribution:** 2 fair
**Rating:** 4
**Confidence:** 3

**Summary:**

This paper proposes a new few-shot learning method leveraging the pre-trained multi-model backbone CLIP. The method aims to eliminate the spurious information within the text embedding, and consequently regularize the image features to deliver better few-shot learning results. To do so, the authors propose to average pooling the text embeddings for the non-contextual information and enforce their KL distance with the image features. The evaluate the proposed method, the authors conduct experiments on multiple benchmarks in different settings.

**Strengths:**

-- The authors’ motivation sounds interesting. It’s a new angle to disentangle the text embeddings and accordingly regularize the image features.

++ The results are looking good, the proposed method achieves good results on multiple benchmarks and settings (1, 2, 4, 8, 16 shots).

**Weaknesses:**

-- Although the motivation is clear and interesting and the authors claim to disentangle the causal information from the spurious information, I didn’t see any clear visualisations to support their claim. In Figure 1, the authors show better class centroids for their method, but it cannot directly show the effect of disentanglement. I suggest the authors show some similarity maps on both learned causal and spurious representations in future versions.

-- For the same purpose, what if employing a small off-the-shell saliency model to first mask the spurious information?

-- This paper can further benefit from improved organization. In Section 2.2, the authors simply list the related works without meaningful discussions. The readers, especially the ones that are not directly working on the same topic would be confused about details such as 1) Is it a common practice to jointly incorporate a frozen and a learnable visual encoder in few-shot learning with CLIP? Is any of the mentioned literature doing so? If not, what’s the gain of this design and what’s the learnable-parameter vs. accuracy trade-off? 2) Is there any changes in the prompt group compared with the literature? Authors claim to identify outliers with K-means. However, prompts like `a toy {c}` and `the plastic {c}` cannot describe most of the images and introduce outliers and it’s easy to simply just remove these manually designed prompts.

-- Also, in section 2.1, the authors mostly introduce the datasets instead of introducing and discussing the few-shot learning literature. Looks like it can be moved to the experiment section instead of related work.

-- In the proposed framework, the authors conduct Isolation forest as well as KMeans to eliminate outlier features, I wonder whether it slows down the training and inference.

-- Minor: the scores for Zero-shot CLIP in tables 1 and 2 are not aligned well.


**Questions:**

See weakness.

---

> ### Author Rebuttal · Authors · 2023-08-09
>
> Q1: Lack of clear visualizations to show how causal information is disentangled from spurious information. Better class centroids do not indicate the effect of disentanglement. Similarity maps on both learned causal and spurious representations are encouraged to be shown.
>
> A1: Figure 1 of the paper visualizes the features of the same category under different domains. In order to better demonstrate the effectiveness of our motivation and the effect of disentanglement, we visualize the features of different categories under different domains. As shown in **Figure 2 in the attached PDF of "global" response**. We employ t-SNE to visualize distinct categories of image features within the contexts of both cartoon and sketch representations. This enables us to estimate the distribution of each category, delineated by an elliptic region. CLIP exhibits the capability to discern nuanced variations across diverse domains and categories. Subsequently, we fine-tuned the model on miniImageNet's training set. On sketch images with large differences in domain from miniImageNet, **the distinction of the feature extracted by fine-tunes model between the two categories deteriorates**. And the **distinction between domains becomes less obvious**. While our method can guarantee the differentiation between categories with larger differences and between domains.
>
> Q2: What if employing a small off-the-shell saliency model to first mask the spurious information?
>
> A2: Thanks for the question. Although the concept of utilizing a saliency model for masking spurious information seems align with image-related tasks, its applicability largely depends on the specific context. **Since the spurious information may be different across tasks, directly using such masking approach may get worse performance**. For instance, image background is usually considered as spurious information and should be masked, while it could also be useful information in some tasks. Paper [1] reports a worse out-of-distribution (OOD) performance when it employs a similar masking approach. In contrast, our method can be applied beyond "object" recognition. In particular, **employing a saliency model on datasets link eurosat is challenging due to the intricate task of satellite image "terrain" classification**. By utilizing non-contextual information from the text and constraining the model's ability to identify spurious information during fine-tuning, we ensure the preservation of model robustness, preventing the degradation of performance.
>
> Q3: Lack of meaningful discussions on related works.
>
> A3: Thanks for the suggestion. Due to space constraints, we have updated the content in global response.
>
> Q4: Is it a common practice to jointly incorporate a frozen and a learnable visual encoder in few-shot learning with CLIP? Is any of the mentioned literature doing so? If not, what’s the gain of this design and what’s the learnable-parameter vs. accuracy trade-off?
>
> A4: In scenarios with ample training samples, a larger number of parameters tends to correlate positively with heightened accuracy. Conversely, in few-shot learning, it's easy to overfit due to too many parameters and not enough data. Consequently, a prevailing trend among existing methodologies is to mitigate the number of learnable parameters. Within the few-shot learning paradigm, enhancing performance through an augmented parameters in the network while concurrently preventing overfitting proves to be challenging. It is worth noting that while existing methods have demonstrated better performance, few have attempted to exploit the potential for fine-tuning the backbone. Accordingly, this paper addresses this challenge by fine-tuning the backbone in few-shot learning  to make it less fitted and improve performance. The experimental results in the following table also demonstrate that applying our fine-tuned backbone directly to existing methods can effectively improve the performance.
>
> |METHOD|1shot|2shot|4shot|8shot|16shot|
> |-|-|-|-|-|-|
> |Tip-Adapter[3]|64.11|64.36|64.63|65.17|65.49|
> |Tip-Adapter+**FD-Align**|64.51|65.33|65.76|66.79|67.28|
> |Tip-Adapter-F[3]|64.64|65.18|65.78|67.21|68.43|
> |Tip-Adapter-F+**FD-Align**|64.86|65.61|66.11|67.58|68.70|
> |APE[4]|65.36|65.69|66.00|66.55|66.55|
> |APE+**FD-Align**|66.71|67.29|67.40|67.76|67.69|
> |APE-T[4]|65.89|66.18|66.82|67.99|68.74|
> |APE-T+**FD-Align**|66.84|67.37|67.81|68.73|69.15|
>
> Q5: Is there any changes in the prompt group compared with the literature? Authors claim to identify outliers with K-means. Why not delete prompts like a toy {c} manually
>
> A5: In order to ensure a fair comparison, we employ the identical prompts as presented in the original CLIP papar.
> It should be corrected that we are using Isolation Forest, not k-means to remove outlies. K-means is employed to mitigate an overabundance of similar prompts, thereby preventing an undue emphasis on a particular context. Since we can't tell the validity of each prompt, there may be cases of mistaken or missed deletions. Consequently, we opt to employ Isolation Forest for the purpose of excluding irrational prompts, as opposed to manual deletion. As an illustration, in Tip-Adapter[3], the prompts after manual filtering by the author contain "itap of a {c}." However, this prompt is not reasonable for our task and leads to performance degradation.
>
> Q6: Related work in Section2.1 and scores in tables 1and 2.
>
> A6: Thanks for your suggestion, we will move it to experiment in the revised version.
>
> Q7: Do Isolation Forest Kmeans slow down training?
>
> A7: No, Isolation Forest and k-means are only run during training initialization. So it doesn't affect training and inference speed.
>
> [1] Masked images are counterfactual samples for robust fine-tuning.
> [2] Conditional Prompt Learning for Vision-Language Models
> [3] Tip-Adapter: Training-free CLIP-Adapter for Better Vision-Language Modeling
> [4] Not All Features Matter: Enhancing Few-shot CLIP with Adaptive Prior Refinement
> [5] Visual Prompt Tuning

---

> > ### Author Response · Authors · 2023-08-15
> > **We would be grateful if you could take a look at the response**
> >
> > Dear Reviewer EKx5,
> >
> > Thank you for reviewing our paper. Just a friendly reminder that **the author-reviewer discussion will close soon**, and we eagerly await your feedback. In response to your comments, we've updated our t-SNE visualization to more clearly highlight our method's advantages and explained how it's different from existing approaches. Could you please take a look at these updates?
> >
> > We're here to discuss any more questions or concerns you may have about our paper.
> >
> > With warm regards,
> >
> > Authors

---

> > > ### Comment · Reviewer_EKx5 · 2023-08-16
> > > **Thanks for the rebuttal**
> > >
> > > I appreciate the efforts the authors spent on their rebuttal! It solves some of my concerns, but I still have two follow-up questions.
> > >
> > > 1. The visualization in the general response still only shows better class centroids instead of the disentanglement. In my understanding, it’s possible to show whether the causal information is disentangled from the spurious information with model explainability techniques such as Grad-CAM [A], [B], or the attention maps. Otherwise, the motivation cannot be fully justified.
> > >
> > > 2. The quantitative results in the A4 Table look good. But since the authors employ two backbones, this may potentially affect the inference speed or GPU memory. I suggest the authors present these efficiency metrics for a better understanding of the accuracy-efficiency trade-off when compared with parameter-efficient fine-tuning methods.
> > >
> > > [A] Selvaraju, R. R., Das, A., Vedantam, R., Cogswell, M., Parikh, D., & Batra, D. (2016). Grad-CAM: Why did you say that?. arXiv preprint arXiv:1611.07450.
> > >
> > > [B] Chefer, H., Gur, S., & Wolf, L. (2021). Transformer interpretability beyond attention visualization. In Proceedings of the IEEE/CVF conference on computer vision and pattern recognition (pp. 782-791).

---

> > > > ### Author Response · Authors · 2023-08-16
> > > > **Further Justification**
> > > >
> > > > Thank you for your valuable feedback. **We humbly acknowledge that our previous rebuttal addressed some of your concerns.** Regarding the points you highlighted about our motivation and efficiency, we hope to offer a more thorough discussion on this matter.
> > > >
> > > > Q1: On Visualization and Motivation
> > > >
> > > > Thank you for your insightful feedback. In light of the existing NeurIPS policy, we are unable to include a direct link at this discussion stage. Nevertheless, we are actively considering petitioning the AC for a special exception. Should both you and the AC agree on the inclusion of a link, **we would promptly provide the image reference**.
> > > >
> > > > Our visualization images compellingly demonstrate that our fine-tuned model emphasizes the object itself more effectively than the directly fine-tuned model. We will ensure that these visualization images are integrated into the revised manuscript for better clarity.
> > > >
> > > > In addition, to some extent, we believe that Figure 1 of our submission and Figure 2 of the global rebuttal also support our motivation. It is worth noting that our decoupling is not a direct decoupling of features. Our goal is to **decouple the ability of the model to obtain both casual and spurious information during the fine-tuning process**, so as to ensure that the fine-tuning does not affect the model's ability to extract spurious information. Figure 1 in our submission shows that our approach achieves **better domain centroids**, i.e., the model retains a **better ability to extract spurious information**. In Figure 2 of the global rebuttal, it can be seen that when fine-tuning is performed directly, the model loses the ability to distinguish between spurious features, and so tends to **confuse unseen out-of-domain data**. The model has **worse class centroids on data with a larger domain gap from the fine-tuned data**, such as sketch data, and our method is able to **preserve the class centroids better**. Therefore, the two figures can partially justify our motivation.
> > > >
> > > > Q2: The efficiency of our method in the inference phase.
> > > >
> > > > Thank you for raising concerns about efficiency. We apologize for any ambiguity in our manuscript. To clarify, during the inference phase, our approach **solely relies on the fine-tuned learnable visual encoder**, which is **structurally identical and has the same parameter count as CLIP's visual encoder**. Consequently, our method does **not impact the inference speed or GPU memory usage** compared to existing methods.
> > > >
> > > > Given that we have addressed part of your concerns and further elaborated on the remaining ones in this response, **we would like to humbly request a reconsideration of the scoring**. We believe our research can provide valuable insights and contributions that would be of great interest to the NeurIPS community.

---

> > > > > ### Author Response · Authors · 2023-08-16
> > > > > **Could we use anonymous links to upload images?**
> > > > >
> > > > > Dear AC,
> > > > >
> > > > > The reviewer would like us to add additional images. Could we use additional anonymous links to add images? Or could we have another way to submit images? Looking forward to your reply.
> > > > >
> > > > > With warm regards,
> > > > >
> > > > > Authors

---

### Official Review · Reviewer_LEKB · 2023-07-27

**Soundness:** 3 good
**Presentation:** 2 fair
**Contribution:** 3 good
**Rating:** 6
**Confidence:** 3

**Summary:**

This paper presents a fine-tuning method for pre-trained models in few-shot learning via CLIP's text and visual feature alignment capability. Specifically, the authors use text information to assist in decoupling spurious information from causal information while keeping the spurious information unchanged during the training process. This way, the authors claim the proposed FD-Align model can maintain its generalizability while effectively mitigating the overfitting problem from previous fine-tuning approaches.

**Strengths:**

The concept of utilizing text to distinguish between causal and spurious feature extraction for enhancing the generalizability of few-shot model training is innovative. Additionally, the discovery that retaining the spurious feature extraction component unchanged pre and post fine-tuning can assist fellow researchers in addressing the few-shot learning issue is noteworthy. In addition to the originality of the concept, the results appear to be encouraging on various datasets when compared to other recent works. Finally, the authors also attached the code for easy reproducibility.

**Weaknesses:**

The presentation and illustrated figures could be improved for better clarity. Additionally, the experimental results do not show comprehensive superiority compared to WiSE-FT, indicating some drawbacks in the proposed approaches.

**Questions:**

According to the authors, keeping the original ability to extract spurious features can aid in solving the few-shot learning problem because it helps reduce overfitting. In situations where the image content is straightforward, identifying the correlation between the background and foreground can be a valuable tool for recognizing unfamiliar objects. For example, an airplane is commonly found in the sky or at an airport rather than in a mountainous environment. However, as image content becomes more complex, it is unclear how much background information is still helpful for this task. Additionally, causal/spurious feature disentanglement appears to be reliant on the quality of the prompt, which can introduce uncertainty into the performance since different individuals may provide different prompts.

**Limitations:**

The authors utilized the underperformed scores to explain the limitation of the proposed method.

---

> ### Author Rebuttal · Authors · 2023-08-09
>
> Q1: The presentation and illustrated figures could be improved for better clarity.
>
> A1: Thanks for your feedback. In response to your suggestion, we have revisited and updated the visual representation of Figure 1. In the modified figure, the data flow and the loss term during the fine-tuning can be shown more clearly. For a more lucid depiction, please refer to **Figure 1 in the attached PDF of "global" response**. We believe the revised figures will offer better clarity and ease of comprehension.
>
> Q2: The experimental results do not show comprehensive superiority compared to WiSE-FT.
>
> A2: Thanks for bringing this to our attention. Upon careful examination, we determined that the minor superiority of our initial experimental results compared to WiSE-FT[2] was due to inadequate hyperparameter tuning. In fact, after re-tuning the hyperparameters of our method, a significant enhancement in performance can be observed. The updated results are shown in the table below.
>
> |      METHOD     |   1 shot  |   2 shot  |   4 shot  |   8 shot  |  16 shot  |
> |:---------------:|:---------:|:---------:|:---------:|:---------:|:---------:|
> |   CLIP(60.33)   |           |           |           |           |           |
> |   LP-CLIP [1]   |   22.17   |   31.90   |   41.20   |   49.52   |   56.13   |
> |   WiSE-FT [2]   |   63.36   |   64.27   |   66.16   |   68.13   |   69.55   |
> | Tip-Adapter [3] |   65.52   |   67.40   |   69.56   |   71.58   |   72.72   |
> |        FT       |   65.24   |   67.42   |   69.85   |   71.73   |   73.00   |
> | **FT+FD-Align** | **65.65** | **68.26** | **71.25** | **73.63** | **75.59** |
>
> Q3: It is unclear how much background information is still helpful as image content becomes more complex?
>
> A3: In few-shot learning, when the content in an image is more complex, it is more difficult for the model to learn the task-relevant causal information and overfit to task-irrelevant spurious information such as background. Our method can effectively avoid the model overfitting to task-irrelevant spurious information. The more complex the background information is, the more effective our method is.
>
> Q4: Causal/spurious feature disentanglement appears to be reliant on the quality of the prompt, which may lead to uncertainty in performance.
>
> A4: Thank you for your insightful question. Indeed, the performance will be influenced by the quality of the prompt. However, we would like to clarify that our method has conscientiously taken steps to mitigate this dependency. Firstly, our approach leverages an effective outlier removal algorithm (i.e., Isolation Forest) to eliminate impractical prompt instances, contributing to a more stable process and diminishing the influence of prompt quality. Secondly, we employ the k-means clustering to reduce redundant prompts, thereby ensuring a representative set of data. When sufficient prompts are available (e.g., large pre-trained language models like GPT can easily offer ample relevant prompts), the above techniques can reduce the impact of prompt quality on performance uncertainty to a minimal level. It's worth emphasizing that **an increase in the number of prompts does not adversely affect training time**, as context weight generation is executed solely prior to the training procedure. We hope this explanation addresses your concern and provides a clear understanding of our approach. We will add the above discussion into the revised paper.
>
> [1] Learning Transferable Visual Models From Natural Language Supervision
>
> [2] Robust fine-tuning of zero-shot models
>
> [3] Tip-Adapter: Training-free CLIP-Adapter for Better Vision-Language Modeling

---

### Author Rebuttal · Authors · 2023-08-09

## Additional discussion in related work.

In the context of few-shot learning, full fine-tuning of the pre-trained model often results in overfitting due to the limited sample size, which consequently diminish the model's generalization ability. Therefore, it is common practice in few-shot learning scenarios to fix the feature extractor while fine-tuning the classification head or training additional structures. For instance, CoOp [1] models prompt’s context words with learnable vectors, keeping the entire pre-trained parameters fixed. Tip-Adapter[2] and APE[3] do not require any back propagation for training the adapter but create the weights through a key-value cache model constructed from the few-shot training set. VPT[4] introduces extra learnable parameters into the input space. However, all these methods are processed with the backbone frozen, while our paper aims to further explore the possibility of fine-tuning the backbone itself.

## Figure
We have updated the method figure and the visualization of the features extracted by the different models in the pdf.

## Table 1

Previous submissions suffered from inadequate hyperparameter tuning. In response, we meticulously re-tuned our hyperparameters to bolster performance, consequently yielding superior results. To provide a comprehensive perspective, we present the latest outcomes in the subsequent table.

|      METHOD     |   1 shot  |   2 shot  |   4 shot  |   8 shot  |  16 shot  |
|:---------------:|:---------:|:---------:|:---------:|:---------:|:---------:|
|   CLIP(60.33)   |           |           |           |           |           |
|     LP-CLIP     |   22.17   |   31.90   |   41.20   |   49.52   |   56.13   |
|     WiSE-FT     |   63.36   |   64.27   |   66.16   |   68.13   |   69.55   |
|   Tip-Adapter   |   65.52   |   67.40   |   69.56   |   71.58   |   72.72   |
|        FT       |   65.24   |   67.42   |   69.85   |   71.73   |   73.00   |
| **FT+FD-Align** | **65.65** | **68.26** | **71.25** | **73.63** | **75.59** |

## Table 2
Performance comparison of replacing our fine-tuned backbone onto an existing method on ImageNet.

|METHOD|1shot|2shot|4shot|8shot|16shot|
|-|-|-|-|-|-|
|Tip-Adapter[2]|64.11|64.36|64.63|65.17|65.49|
|Tip-Adapter+**FD-Align**|64.51|65.33|65.76|66.79|67.28|
|Tip-Adapter-F[2]|64.64|65.18|65.78|67.21|68.43|
|Tip-Adapter-F+**FD-Align**|64.86|65.61|66.11|67.58|68.70|
|APE[3]|65.36|65.69|66.00|66.55|66.55|
|APE+**FD-Align**|66.71|67.29|67.40|67.76|67.69|
|APE-T[3]|65.89|66.18|66.82|67.99|68.74|
|APE-T+**FD-Align**|66.84|67.37|67.81|68.73|69.15|

## Table 3
A comparison of the performance of replacing our fine-tuned backbone on the existing method in the OOD task. The backone is fine-tuned on 16 shots imageNet.

|           Method           | imageNet | imageNetA | imageNetR | imageNetS | imageNetV2 |
|:--------------------------:|:--------:|:---------:|:---------:|:---------:|:----------:|
|            CLIP            |   63.34  |   31.57   |   68.45   |   42.31   |    55.92   |
|             FT             |   64.91  |   30.05   |    68.7   |   42.24   |    57.63   |
|       FT+**FD-Align**      |   66.39  |    31.8   |    69.7   |    43.5   |    57.73   |
|         Tip-adapter        |   65.49  |     -     |     -     |   42.48   |    57.58   |
|  Tip-adapter+**FD-Align**  |   65.49  |     -     |     -     |   43.84   |    59.10   |
|        Tip-adapter-F       |   68.43  |     -     |     -     |   42.54   |    59.58   |
| Tip-adapter-F+**FD-Align** |   68.70  |     -     |     -     |   43.67   |    60.17   |
|             APE            |   66.55  |     -     |     -     |   43.28   |    58.31   |
|      APE+**FD-Align**      |   67.69  |     -     |     -     |   44.23   |    59.36   |
|            APE-T           |   68.74  |     -     |     -     |   43.23   |    59.58   |
|     APE-T+**FD-Align**     |   69.15  |     -     |     -     |   44.04   |    60.83   |

[1] Conditional Prompt Learning for Vision-Language Models

[2] Tip-Adapter: Training-free CLIP-Adapter for Better Vision-Language Modeling

[3] Not All Features Matter: Enhancing Few-shot CLIP with Adaptive Prior Refinement

[4] Visual Prompt Tuning

---

### Author Response · Authors · 2023-08-14
**Seeking Your Further Feedback**

Dear AC and Reviewers,

We would like to express our sincere gratitude for the time and effort you have dedicated to the evaluation of our work. Your valuable insights and expertise are deeply appreciated.

In response to your review comments, we have prepared detailed answers to address the concerns and inquiries you have regarding our paper. **Should there be any unresolved issues, or should you need further clarification or have additional questions, please do not hesitate to let us know.** We stand ready to provide any information or clarification that may assist you in your review.

Thank you once again for your invaluable time and consideration. We eagerly look forward to hearing from you, and we hope to have the opportunity to respond to any feedback or questions you may have.

With warm regards,

Authors

---

### Decision · Program_Chairs · 2023-09-21

**Decision:**

Accept (poster)

**Comment:**

The paper received quite diverse ratings. After a thorough examination of the paper, reviews, authors' rebuttals, and the discussion, the AC has identified the following key points of contention:

Motivation and Its Adequacy: Reviewers EKx5, ePYg, and Teuf initially raised concerns about the clarity and sufficiency of the paper's motivation. However, the AC concur with the authors that the idea of "distinguishing between causal information and spurious information" is well-motivated and substantiated by Figures 1 and 2. Reviewers EKx5 and ePYg also acknowledge the motivation as clear and interesting. The authors' final response to Reviewer Teuf appears to have addressed the concerns adequately.

Novelty of the Approach: Reviewer vxTm questioned the novelty of the proposed method. The AC agrees with the authors that the concept of "utilizing text to distinguish between causal and spurious feature extraction for enhancing the generalizability of few-shot model training is innovative." However, it is essential for the authors to emphasize this concept in the Introduction to better convey the innovation to other researchers.

Insufficient Experimentation: The issue of insufficient experiments has been raised on several fronts:
a) Reviewer EKx5 inquired about the efficiency of the proposed method and its impact on inference speed. It appears that the authors' final response adequately addressed this concern.
b) Reviewer vxTm expressed concerns about the limited performance improvements demonstrated in the experiments. The authors' response seems to provide a solution to this issue. The AC strongly recommends that the authors revise the paper by including the updated results and offering a detailed explanation of how these results were achieved.

c) Reviewer Teuf suggested the inclusion of experiments with different backbones, such as ResNet50 and ViT-B/16, to provide a more comprehensive performance comparison. While this is a valuable suggestion to enhance the paper, it does not constitute a reason for rejection.

Overall, considering the paper's contributions in terms of novelty, methodology, and experiments, the AC believes it meets the criteria for acceptance at NeurIPS. However, the AC recommends that the authors make the suggested revisions to further strengthen the paper and address the concerns raised by the reviewers.